# Structural basis for specific inhibition of the deubiquitinase UCHL1

Christian Grethe [1,2,4], Mirko Schmidt [1,2,4], Gian-Marvin Kipka[1,2], Rachel O'Dea[1,2], Kai Gallant[1,2], Petra Janning[3] & Malte Gersch [1,2] ✉

Ubiquitination regulates protein homeostasis and is tightly controlled by deubiquitinases (DUBs). Loss of the DUB UCHL1 leads to neurodegeneration, and its dysregulation promotes cancer metastasis and invasiveness. Small molecule probes for UCHL1 and DUBs in general could help investigate their function, yet specific inhibitors and structural information are rare. Here we report the potent and non-toxic chemogenomic pair of activity-based probes GK13S and GK16S for UCHL1. Biochemical characterization of GK13S demonstrates its stereoselective inhibition of cellular UCHL1. The crystal structure of UCHL1 in complex with GK13S shows the enzyme locked in a hybrid conformation of apo and Ubiquitin-bound states, which underlies its UCHL1-specificity within the UCH DUB family. Phenocopying a reported inactivating mutation of UCHL1 in mice, GK13S, but not GK16S, leads to reduced levels of monoubiquitin in a human glioblastoma cell line. Collectively, we introduce a set of structurally characterized, chemogenomic probes suitable for the cellular investigation of UCHL1.

Ubiquitination acts as a highly versatile posttranslational modification in cells which can be reverted by deubiquitinases (DUBs)[1]. Humans possess about 100 of these specialized hydrolases, which cleave the isopeptide bond between the Ubiquitin carboxy terminus and the lysine sidechain of a substrate. Through counteracting the activity of E3 ligases, DUBs control the strength of Ubiquitin-dependent signaling in cells and are therefore critical regulatory components of the Ubiquitin system[2].

The Ubiquitin C-terminal hydrolase (UCH) family of DUBs comprises four members, of which UCHL1 is the smallest and one of the most studied deubiquitinases[3]. UCHL1 is highly expressed in brain tissue, where it can make up to 5% of total soluble brain protein[3], and its detection in blood has received FDA-approval for diagnosis of intracranial lesions. While dispensable for neuronal development, UCHL1 is strictly required for axonal maintenance and integrity[3]. Mutations in the UCHL1 gene in humans lead to progressive early onset neurodegeneration[4]. In mice, mutations of

UCHL1 are associated with gracile axonal dystrophy[5] and lead to reduced levels of free monoubiquitin in brain tissue[6,7]. Complete loss of UCHL1 in mice causes impaired synaptic transmission at neuromuscular junctions and denervation of muscles[8]. In addition, the expression of UCHL1 was shown to be dysregulated in various tumor entities including pancreatic cancer, colorectal cancer, and breast cancer, where it can promote cancer invasion and metastasis[9,10]. A myriad of other functions of UCHL1 involving the control of metabolism, protein aggregation, and autophagy have been reported[3].

Enigmatically, recombinant UCHL1 does not cleave Ubiquitin chains and has very limited activity against folded ubiquitinated substrates[11]. This behavior has been structurally rationalized by its crossover loop, which spatially restricts access of substrates to its active site cysteine and which is the smallest within the UCH family of DUBs[12–14]. Hence in addition to the deubiquitination of small peptides[15], additional functions of UCHL1 involving Ubiquitin binding[6], Ubiquitin

[1]Max Planck Institute of Molecular Physiology, Chemical Genomics Centre, Otto-Hahn-Str. 15, Dortmund, Germany. [2]TU Dortmund University, Department of Chemistry and Chemical Biology, Otto-Hahn-Str. 15, Dortmund, Germany. [3]Max Planck Institute of Molecular Physiology, Department of Chemical Biology, Otto-Hahn-Str. 11, Dortmund, Germany. [4]These authors contributed equally: Christian Grethe, Mirko Schmidt. ✉e-mail: malte.gersch@mpi-dortmund.mpg.de

conjugation[16], neuronal antioxidation[17], and transnitrosylation[18] have been described.

Specific small molecule inhibitors of UCHL1 have the potential to aid in the mechanistic investigation of its enzymatic function[19,20], yet few DUB inhibitors are known, with many of unknown or relatively poor specificity[21]. Moreover, rational endeavors to improve the potency and specificity of DUB inhibitors are often hampered by the lack of structural information from specific inhibitor:DUB complexes[22,23] as is also the case for UCHL1. Following the publication of 2- and 3-carboxy-*N*-cyanopyrrolidines as UCHL1 inhibitors in the patent literature[24,25], the Flaherty[26], Tate[27,28], and Ovaa/Geurink[29] labs published small molecule activity-based probes for UCHL1. These showed highly potent target engagement in cell lines, covalent inhibition in vitro, and enabled the visualization of UCHL1 activity in live zebrafish. However, some reported compounds showed pronounced cytotoxicity, which is at odds with the viability of many UCHL1 knockout cell lines[30]. Moreover, various other covalently bound

targets were identified which could not be disentangled with the used set of control probes[28]. In addition, no phenotype in protein abundance following UCHL1 inhibition was detected[28].

Here we report the potent and nontoxic UCHL1-targeting 3-carboxy-*N*-cyanopyrrolidine activity-based probe GK13S from a screen of candidate DUB-targeting nitriles. The crystal structure of UCHL1 in complex with the probe shows UCHL1 locked in a hybrid conformation with some residues in the UCHL1-specific apo and others in the Ubiquitin-bound state. The structure reveals the basis for the inhibitory potency of the probe as well as its exquisite specificity within the highly homologous UCH family of DUBs. We describe how GK13S, together with GK16S, forms a chemogenomic pair of probes which enable the specific investigation of UCHL1 function in cells. Phenocopying the effect that an inactivating mutation of UCHL1 has on the mouse brain, GK13S, but not GK16S, led to the reduction of monoubiquitin levels in the human glioblastoma cell line U-87 MG. Collectively, we introduce a set of

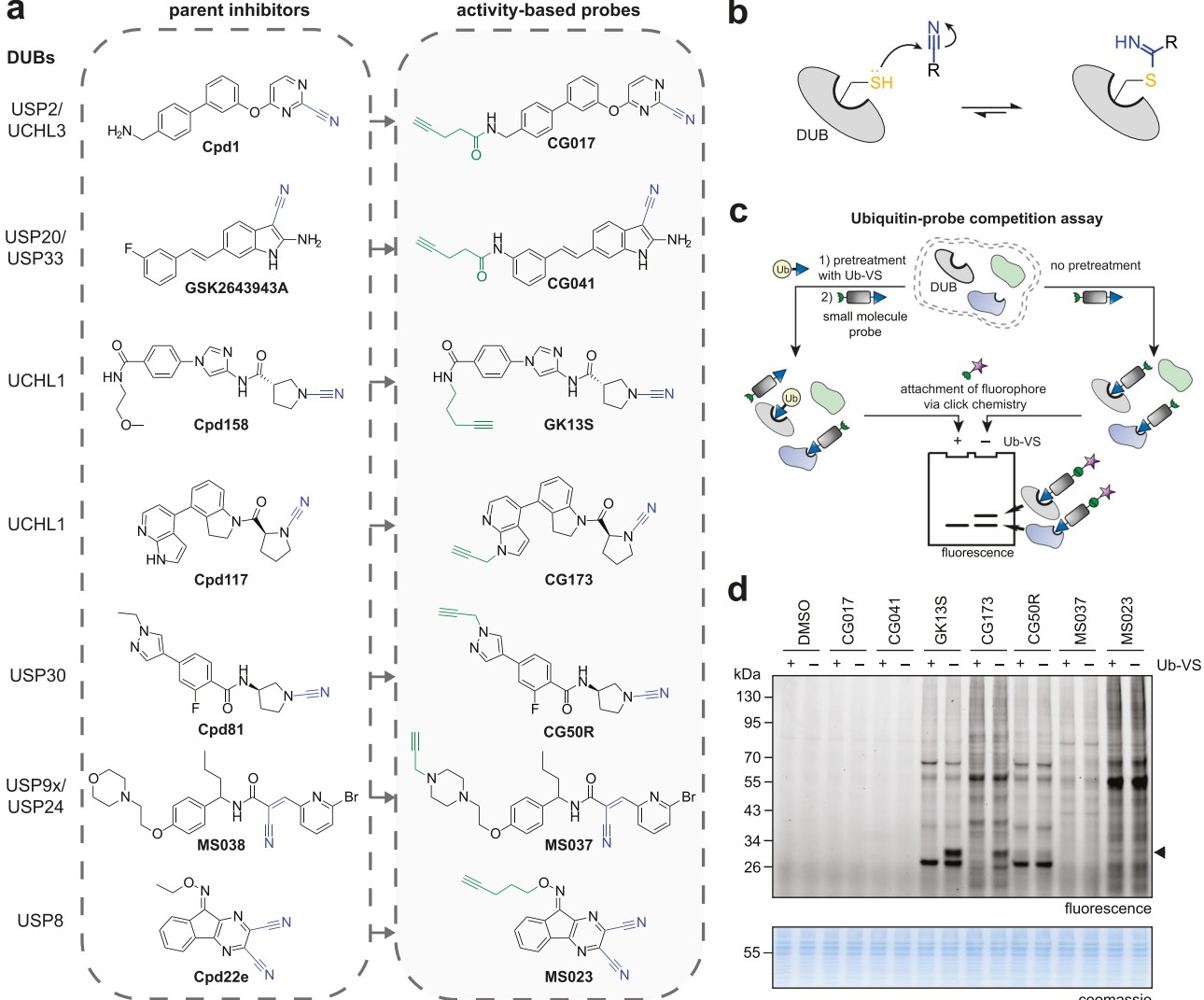

**Fig. 1 | Design and evaluation of an activity-based probe library derived from nitrile-based DUB inhibitors. a** Synthesized activity-based probes (right panel) derived from known inhibitors (left panel), containing a potential nitrile-based warhead (blue) for covalent protein modification and an alkyne handle (green) for bioorthogonal functionalization. See the supporting information for chemical synthesis routes. **b** Mechanism of thioimidate formation between DUB active site cysteine and small molecule nitrile. **c** Ubiquitin vinyl sulfone (Ub-VS) competition workflow. Cell lysates were either incubated with a Ub-VS probe followed by small molecule probe treatment (left path) or treated with the small molecule probe only (right path), in order to indicate whether small molecule probe-modified proteins are DUBs. **d** Activity-based protein profiling in HEK293 cell lysate with indicated compounds (1 μM, 1 h). Proteomes were pretreated with Ub-VS probe where shown. The black arrow indicates a Ub-VS competitive protein target in GK13S- and CG173-treated samples. Uncropped versions of gels are shown in the supplementary information.

structurally characterized, chemogenomic probes suitable for the cellular investigation of UCHL1 function.

## Results

### A Ubiquitin-probe competition workflow identifies DUB-binding nitriles

Aliphatic nitriles, aryl nitriles and cyanamides of diverse scaffolds have been described as potent covalent inhibitors for various enzymes (Supplementary Fig. 1). We also noted that multiple nitrile-featuring inhibitors for deubiquitinases of the USP and UCH families have been reported, but are generally poorly characterized with regards to their mode of inhibition and their specificity[24,25,31–35]. In an effort to characterize these, we selected a panel of seven structurally diverse parent inhibitors and synthesized corresponding alkyne-tagged activity-based probes[36] (Fig. 1a and Supplementary Information). We reasoned that these may form covalent thioimidate conjugates with the active site cysteine of deubiquitinases (Fig. 1b) and possibly other cellular proteins and may thus report on their covalent targets in an unbiased manner. In order to evaluate whether these compounds bind deubiquitinases in a complex proteome, we used a Ubiquitin probe[37] competition experiment (Fig. 1c). HEK293 lysate was either treated directly with the compounds or pretreated with the DUB-reactive probe Ubiquitin vinyl sulfone (Ub-VS) followed by small molecule probes. Compound-bound proteins were then visualized through copper-catalyzed click chemistry and samples were analyzed by SDS-PAGE (Fig. 1d). We noted a Ubiquitin-probe competitive band of approx. 30 kDa for the 2- and 3-carboxypyrrolidine probes GK13S and CG173, respectively, which are derived from inhibitors of the DUB UCHL1. This band was also observed in PC-3 cell proteome but absent in proteome from HeLa and MCF-7 cells which are known to have a very low UCHL1 expression level[9,10] (Supplementary Fig. 2a–c). Other non-Ubiquitin-probe competitive bands, including a pronounced one with lower molecular weight, indicated additional non-DUB targets for these compounds. Despite the presence of a large number of active DUBs in these sampled cell lines[38], no other Ubiquitin-probe-competitive bands were observed. CG50R, MS037 and MS023 did not show DUB engagement in this assay, whereas CG017 and CG041 did not show any covalent-irreversibly bound proteins at all.

### A set of chemogenomic probes for UCHL1

Activity-based probes generally consist of a warhead for covalent target engagement, a specificity element directing target selection and a handle for bioorthogonal functionalization[36,39]. We focused on compound GK13S, which showed the strongest DUB-probe competitive signal in lysate, and synthesized a systematic set of compounds comprising probes of both stereoisomers (GK13S and GK13R), minimal probes lacking the central aromatic specificity element (GK16S and GK16R) as well as inactive controls lacking the warhead (GK12S and GK12R) (Fig. 2a and Supplementary Information). To assess the specificity of these probes, we treated intact HEK293 cells and visualized covalently bound proteins after 24 h through activity-based profiling (Fig. 2b). While at the high concentration of 10 µM all probes showed binding to a lower molecular weight protein, at 0.1 and 1 µM only GK13S showed a strong signal for a band of ~30 kDa. To confirm that this protein corresponds to the Ubiquitin-probe competitive band observed in lysate, we carried out a reverse Ubiquitin-probe competition experiment. Here intact cells were treated with compounds, their lysate was then incubated with the HA-Ub-VS probe, and active DUBs were visualized by anti-HA western blotting (Fig. 2c). The molecular weight and the strong signal decrease in the HA-Ub-VS competition experiment were consistent with UCHL1 which is the DUB target of the parent inhibitor. As we were concerned about reversibility and incomplete protein modification, we had optimized all steps and identified a click chemistry condition in which recombinantly purified UCHL1 showed near complete modification and established that the

protein-probe isothiourea linkage was stable under denaturing conditions (Supplementary Fig. 2d, e). The near complete shift of UCHL1 in the western blot of the GK13S-treated samples (Fig. 2b) suggested that a high degree of modification could be reached in cells.

To identify probe-bound proteins, we used an activity-based protein profiling workflow involving the enrichment of bound proteins via streptavidin and protein analysis by quantitative mass spectrometry (Supplementary Fig. 3a)[39]. UCHL1 was the most enriched protein from cells treated with 5 µM of GK13S when compared to DMSO (Fig. 2d), with a logarithmic enrichment factor of 11.5 (equivalent to ~3000-fold) which is in line with the strong signal on the gel (Fig. 2b). At this higher concentration, GK13S also led to the strong enrichment of PARK7 (also known as DJ-1) and the PARK7-homolog C21orf33 (also known as GATD3, a mitochondrial glutamine amidotransferase), as well as several aldehyde dehydrogenases and the Omega-amidase NIT2. Moreover, a large number of weakly enriched proteins was detected, which is consistent with the background observed in the gel-based labeling assay (Fig. 2b). At a lower concentration of 1 µM, UCHL1 (with an enrichment factor of 8), PARK7 and C21orf33 were still strongly enriched. Importantly, both PARK7 and C21orf33, but not UCHL1, were also strongly enriched by the minimal probe GK16S, which bound cellular UCHL1 with an enrichment factor of only 3 (Fig. 2e).

We next validated targets through genetic perturbation. siRNA-mediated depletion led to the unambiguous assignment of the upper band labeled by GK13S as UCHL1 and the lower band labeled by both GK13S and GK16R as PARK7 (Fig. 2f). This experiment also clarified that the faint band observed in GK16S-treated samples above PARK7 is not UCHL1, as its intensity is unchanged in the UCHL1-depleted sample while it is consistent with the molecular weight of C21orf33. Overexpression of wild-type UCHL1, but not of its catalytically inactive C90A mutant, confirmed the active center of UCHL1 as the site of labeling by GK13S (Fig. 2g). Likewise, overexpression of wild-type PARK7, but not its catalytic cysteine mutant C160A, led to labeling by all probes tested (Fig. 2h).

Chemogenomic probes denote well-characterized tool compounds for the functional annotation of enzymes in cellular systems[40]. Where a probe which is specific for only a single protein is not available, a comparison of two chemogenomic probes may allow for the specific investigation of a non-overlapping protein target. The comparison of GK13S to GK16S-enriched proteins (Fig. 2e, right panel) suggested that GK13S binds all targets of GK16S but also UCHL1 in addition. A similar pattern was observed for the enantiomer GK13R, which in agreement with the labeling experiment (see Fig. 2b), showed a lower enrichment of UCHL1 than GK13S when compared to either DMSO or the minimal probe (Supplementary Fig. 3b). Collectively, these results suggested that GK13S and GK16S are such a set of chemogenomic probes suitable for the small molecule-mediated investigation of cellular functions of UCHL1.

### GK13S potently inhibits recombinant and cellular UCHL1

To characterize the specificity of GK13S for UCHL1 more thoroughly, we analyzed binding and inhibition with recombinant protein. At a probe concentration of 1 µM, GK13S, but not its enantiomer, bound UCHL1 in 1:1 stoichiometry as determined by intact protein mass spectrometry (Fig. 3a). $IC_{50}$ measurements at a fixed incubation time of 1 h revealed that the probe and the parent inhibitor have comparable potencies of 50 and 129 nM, respectively, whereas the stereoisomer inhibited about 40-fold worse. Probes lacking the warhead as well as minimal probes did not show any appreciable degree of inhibition ($IC_{50} > 100$ µM), highlighting the need for both the warhead as well as the aromatic specificity element in GK13S for UCHL1 inhibition (Fig. 3b). These data are in agreement with the labeling experiments (Fig. 2b, c) as well as protein mass spectrometry measurements at higher compound concentrations (Supplementary Fig. 4a, b). The

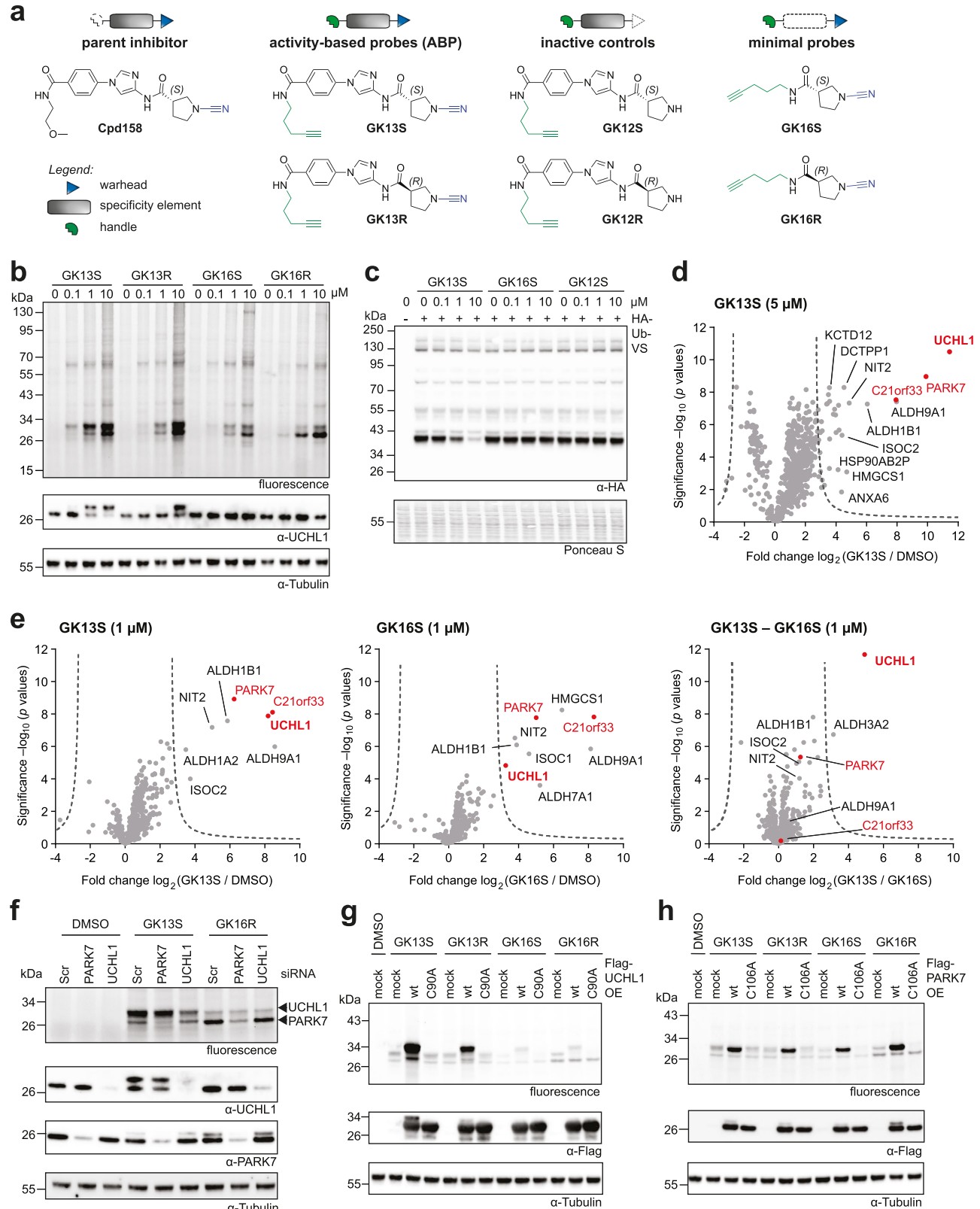

covalent mode of inhibition was further characterized with a $k_{obs}/[I]$ value of GK13S of 681 $M^{-1}s^{-1}$ (Fig. 3c), which is in line with the relatively weak electrophilicity of the cyanamide and comparable to other warheads[41], but still approximately 40-fold larger than for the enantiomer (Supplementary Fig. 5a, b). The binding mode was characterized as covalent-irreversible as inhibition was sustained in a jump-dilution assay (Supplementary Fig. 4c, d), and UCHL1 remained bound to GK13S even after unfolding in 5 M urea and subsequent dialysis (Supplementary Fig. 4e). Consistent with specific recognition of the probe by UCHL1, both the probe as well as the inhibitor, but not the other compounds, led to an increase in protein stability of 6 °C (Fig. 3d). To assess the degree of inhibition of cellular UCHL1, we

**Fig. 2 | A set of chemogenomic probes for UCHL1 and PARK7 from 1,3-linked cyanopyrrolidines. a** Schematic representations and chemical structures of synthesized probes and controls comprising warhead (blue), specificity element (gray) and alkyne handle (green). **b** Cellular activity-based protein profiling with intact HEK293 cells and indicated compounds (24 h incubation). The fluorescence image visualizes probe-bound endogenous proteins. **c** Cellular HA-Ub-VS competition experiment. HEK293 cells were treated with indicated compounds for 24 h. Lysates were then incubated with an HA-Ub-VS probe, followed by western blotting visualizing Ub-VS-reactive DUBs. **d, e** Proteomics-based target identification of indicated probes. Volcano plots show the relative label-free abundance ratio (fold change) of proteins between compound-treated cells and DMSO-treated controls

(**d, e**) or between GK13S and the minimal probe GK16S (**e**, right panel). Cells were treated for 24 h at indicated concentrations. UCHL1, PARK7, and PARK7-homolog C21orf33 are marked in red. See Supplementary Fig. 3a for the workflow used. Source data are provided as a Source Data file. **f** Target validation through siRNA-based knockdown of UCHL1 and PARK7 in HEK293 cells after treatment with indicated compounds (1 μM, 1 h). Fluorescence gel band identities of UCHL1 and PARK7 derived from western blots are shown as black arrows. **g, h** Determination of sites of covalent modification through overexpression (OE) of Flag-UCHL1 (wt and C90A active site mutant, **g**) and Flag-PARK7 (wt and C106A active site mutant, **h**) in HEK293 cells after treatment with indicated compounds (1 μM, 1 h). wt, wildtype. Uncropped versions of gels and blots are shown in the supplementary information.

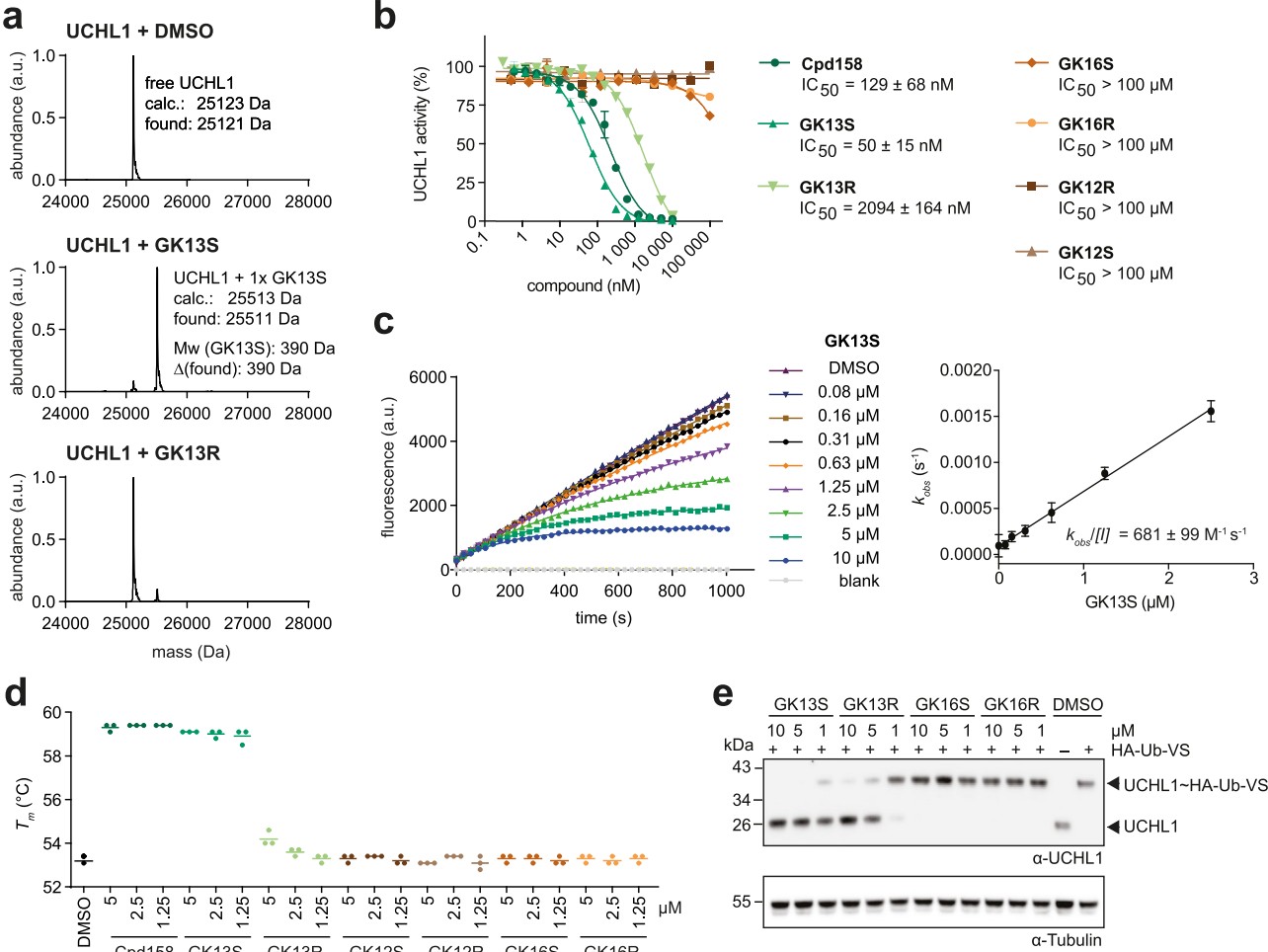

**Fig. 3 | GK13S potently inhibits recombinant and cellular UCHL1. a** Intact protein mass spectrometry revealed covalent binding of GK13S and GK13R to recombinant UCHL1. UCHL1 (0.8 μM) was treated with compound (1 μM) or DMSO for 2 h. **b** Inhibitory potencies of indicated compounds, preincubated with UCHL1 for 1 h, determined from a Ubiquitin rhodamine cleavage assay. Data points are shown as mean ± standard deviation ($N = 2$) from independent experiments. IC$_{50}$ values were determined from 5 (Cpd158), 3 (GK13S), or 2 (GK13R) independent experiments. Source data are provided as a Source Data file. **c** $k_{obs}/[I]$ kinetic assay of GK13S binding to UCHL1 at indicated concentrations. Data points are shown as means calculated from $N = 3$ wells ($N = 6$ wells for DMSO and blank (i.e., Ubiquitin rhodamine without enzyme)). Rate constants $k_{obs}$ were determined from the plot on the

left, and then plotted against inhibitor concentrations as shown on the right for a representative experiment. The $k_{obs}/[I]$ value was determined as the slope and is given as means ± standard deviation calculated from five independent experiments. Source data are provided as a Source Data file. **d** Thermal shift assay showing the melting temperature ($T_m$) of UCHL1 (1 μM) pretreated for 1 h with compounds at indicated concentrations. Source data are provided as a Source Data file. **e** Inhibition of cellular UCHL1. Western blot analysis of endogenous UCHL1 labeled with HA-Ub-VS after treatment of HEK293 cells with either the indicated compounds or DMSO for 24 h. Uncropped versions of gels and blots are shown in the supplementary information.

employed a Ub-VS-mediated target engagement assay[38]. Following treatment of HEK293 cells for 24 h with 1 μM of GK13S, near complete inhibition of UCHL1 was observed, whereas other compounds, including the minimal probes, did not inhibit UCHL1 under these conditions. Full inhibition was achieved with 5 μM (Fig. 3e). Since

covalent inhibitors act in a time-dependent manner, it is reasonable that the rather large specificity window of GK13S vs GK16S observed in the in vitro assay with 1 h incubation (50 nM vs >100 μM) is reduced when a cellular inhibition with 24 h incubation is assessed (change in log2 enrichment values of 8−3 = 5, equating to >32 fold difference).

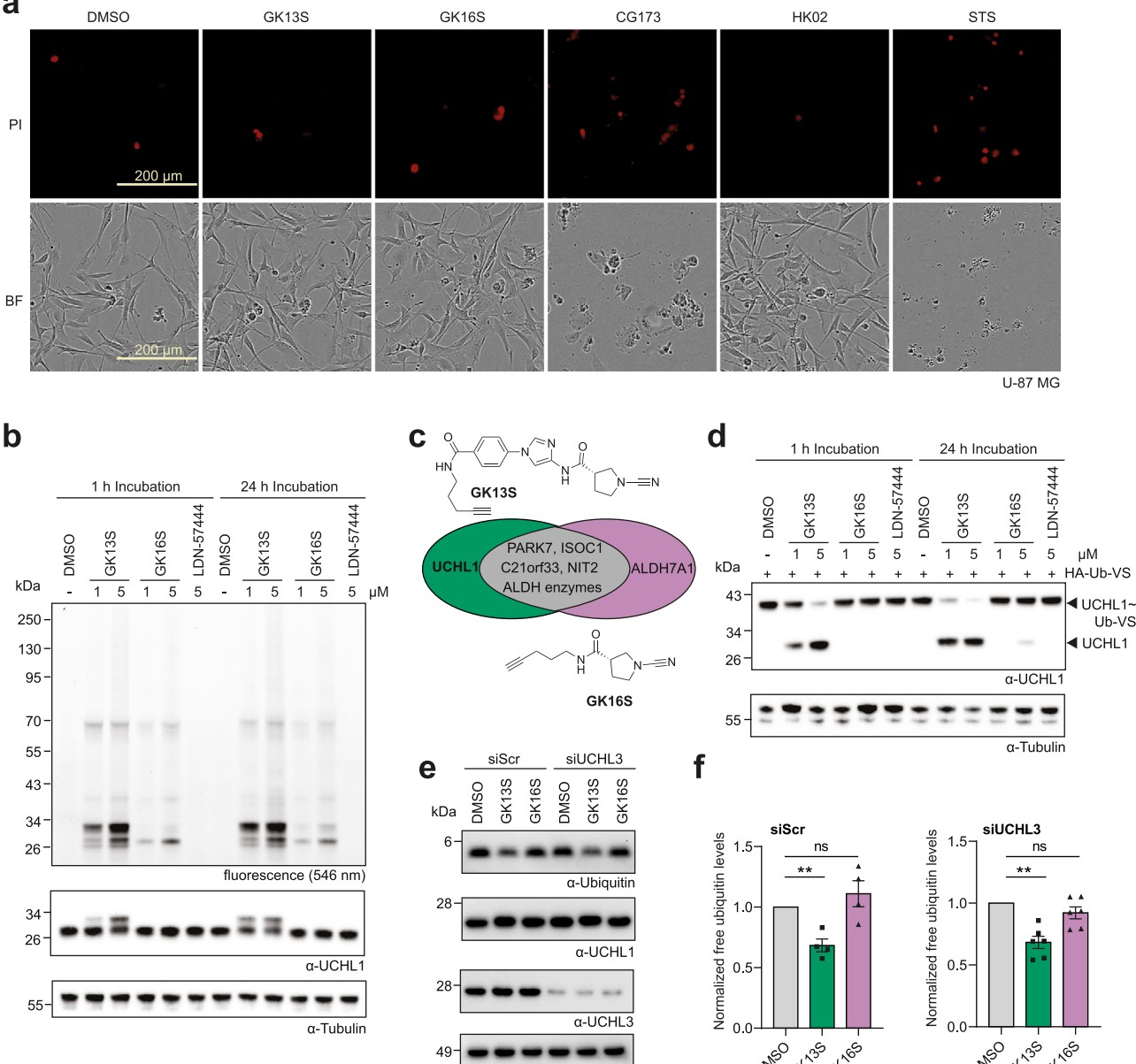

**Fig. 4 | GK13S, but not GK16S, reduces monoubiquitin in U-87 MG cells and thereby phenocopies the effect of a UCHL1 mutant mouse. a** Representative microscopy images of U-87 MG cells were taken 72 h after treatment with 1.25 μM of the indicated compounds (including the control Staurosporine, STS). Effects of the compounds on cell growth and viability were assessed using analysis of confluency (bright-field, BF) and apoptosis (propidium iodide staining, PI). See Supplementary Fig. 7a–d for quantitation. **b** Cellular activity-based protein profiling of intact U-87 MG cells treated with indicated compounds or DMSO for 1 or 24 h. **c** Schematic representation of overlapping and individual cellular targets of GK13S and GK16S, supporting their application as chemogenomic probes for the investigation of UCHL1. Compare Fig. 2e. **d** Inhibition of cellular UCHL1. Western blot analysis of endogenous UCHL1 labeled with HA-Ub-VS after treatment of U-87 MG cells with either the indicated compounds or DMSO for 1 or 24 h. **e** Western blots showing

monoubiquitin levels in U-87 MG cells treated with GK13S or GK16S. An additional knockdown of UCHL3 does not aggravate the effect. **f** Quantification of mono-ubiquitin intensities upon treatment of U-87 MG cells with DMSO, GK13S, or GK16S (as shown in **e**). Values correspond to the mean of four or six independent experiments for control or UCHL3 knockdown, respectively. Error values represent the standard error of the mean (s.e.m.). Statistical significance was analyzed using individual one-sample, two-tailed *t*-tests compared to the mean of "1" as set for the DMSO-treated samples. **$p < 0.01$ (exact $p = 0.0095$ for DMSO/GK13S comparison in siScr background and $p = 0.0014$ for the DMSO/GK13S comparison in the siUCHL3 background); ns, not significant. Uncropped versions of gels and blots are shown in the supplementary information. Source data are provided as a Source Data file.

This led to the small, but significant detection of UCHL1 by GK16S in the pulldown experiment (Fig. 2e). However, this amount did not equate to a detectable inhibition of the cellular UCHL1 population (Fig. 3e). Consistent with this notion, the in vitro potency difference of GK13S and its enantiomer GK13R (50 nM vs. 2 μM, ~40-fold) did not translate into a similarly wide potency window in a cellular context (change in log2 enrichment values of 2, equating to fourfold difference) (Fig. 3e and Supplementary Fig. 3b). This supports the use of the

minimal probe GK16S, but not of the enantiomer GK13R, as a chemogenomic control for GK13S to investigate the specific inhibition of UCHL1.

## Inhibition of UCHL1 by GK13S does not impair cell growth
As both the 3-carboxypyrrolidine GK13S as well as the 2-carboxypyrrolidine CG173 probes indicated binding to UCHL1 (Fig. 1d), we next compared these compounds in vitro (Supplementary

Fig. 5a–e). CG173 displayed a virtually identical inhibitory potency ($k_{obs}/[I]$ value) as GK13S, with an even tenfold lower IC$_{50}$, however, led to reduced protein stabilization (Supplementary Fig. 5c, d), suggesting less tight binding and a potentially more reactivity-driven inhibition. In line with the latter, we found that CG173 bound UCHL1 covalently in 2:1 stoichiometry (Supplementary Fig. 5e) under conditions where an excess of GK13S bound only once (Supplementary Fig. 4a). In cells, notably, GK13S, as well as the 3-carboxypyrrolidine parental inhibitor Cpd158[25], showed a time- and concentration-dependent inhibition, whereas CG173 and its 2-carboxypyrrolidine parental inhibitor Cpd117[24] showed a very low degree of inhibition after 1 h, and no inhibition of cellular UCHL1 at all after 24 h incubation (Supplementary Fig. 5f). Instead, CG173 showed strong induction of apoptosis and growth arrest of HEK293 cells, at concentrations of 1.3 and 5 μM, on par with the control staurosporine. In contrast, neither GK13S nor GK16S showed growth arrest nor apoptosis of HEK293 cells at concentrations up to 5 μM and an incubation time of up to 72 h (Supplementary Fig. 6a–e). These data are consistent with the viability of a UCHL1 knockout in many cell lines[30] and point to non-UCHL1-related toxicity associated with CG173. As such, the results collectively suggest that GK13S, but not CG173, is suitable for the characterization of UCHL1 function in cells.

## Glioblastoma cells treated with GK13S but not GK16S phenocopy a UCHL1 mutant mouse

UCHL1 is highly expressed in brain tissue[3], and mutations in the UCHL1 gene in humans lead to progressive early onset neurodegeneration[4]. In mice, mutation of UCHL1 is associated with gracile axonal dystrophy[5] and leads to reduced levels of free monoubiquitin in brain tissue[6,7]. This effect has previously been studied through the overexpression of human UCHL1 in the monkey cell line COS-7[42]. We sought to find a human cellular system that recapitulated this phenotype and that could validate the application of our pair of probes. During this search, we focused on the human glioblastoma cell line U-87 MG. We first verified that both GK13S and GK16S are nontoxic in concentrations of up to 5 μM and up to 72 h, as seen for HEK293 cells (Fig. 4a and Supplementary Fig. 7a–d). Next, we established that also in this cell line, GK13S and GK16S could be used as chemogenomic probes (Fig. 4b–d). The pattern of bound proteins in U-87 MG was highly similar to that observed in HEK293 (Fig. 4b). We included the isatin O-acyl oxime LDN-57444[43], which has been widely used as a specific UCHL1 inhibitor, whose effectiveness, however, has recently been questioned[27,29]. We found that GK13S, but neither GK16S nor LDN-57444, led to complete inhibition of cellular UCHL1 in U-87 MG cells (Fig. 4d). To investigate if GK13S and GK16S can be used to assess the consequences of UCHL1 inhibition, we quantified the levels of free monoubiquitin by western blot. Consistently, we found Ubiquitin levels in U-87 MG cells to be reduced after incubation with GK13S, but not with GK16S (Fig. 4e, f). This reduction was unchanged in the background of siRNA-mediated depletion of PARK7 (Supplementary Fig. 7e, f) or UCHL3 (Fig. 4e, f), in line with the presumed non-redundant function of these two homologous DUBs[3]. Collectively, these data show that inhibition of UCHL1 by GK13S in the glioblastoma cell line U-87 MG on a molecular level phenocopies a pathogenic UCHL1 mutation in mice[6]. They further validate the application of the chemogenomic pair of probes to investigate the function of UCHL1 in a cellular context.

## A compound-induced hybrid conformation underlies GK13S-mediated inhibition of UCHL1

We next sought to structurally characterize the binding of GK13S to UCHL1 in order to reveal the basis for its specificity[44], but various soaking and co-crystallization efforts with full-length human UCHL1 were not successful. As both N-and C-terminal truncation leads to protein destabilization[45,46], we applied lysine methylation as a crystallization rescue strategy[47] to drive the formation of a crystal form which

**Table 1 | Data collection and refinement statistics**

| | UCHL1$^{methylated}$~GK13S (PDB code: 7ZMO) |
|---|---|
| **Data collection** | |
| Beamline | SLS – PXII |
| Wavelength | 1.000 Å |
| Space group | $P\,2_12_12_1$ |
| Cell dimensions | |
| $a, b, c$ (Å) | 101.93, 144.41, 158.25 |
| $\alpha, \beta, \gamma$ (°) | 90, 90, 90 |
| Anisotropy correction | yes |
| Observed reflections | 421,721 |
| Unique reflections | 63,622 |
| Resolution (Å) | 62.50 – 2.24 (2.54 – 2.24) |
| Ellipsoidal resolution limits (Å) [direction] | 3.22 [a*] 2.70 [b*] 2.20 [c*] |
| $R_{merge}$ | 0.053 (0.457) |
| $R_{meas}$ | 0.057 (0.499) |
| $I/\sigma(I)$ | 18.0 (3.4) |
| $CC_{1/2}$ | 1.000 (0.917) |
| Spherical Completeness (%) | 56.3 (10.7) |
| Ellipsoidal Completeness (%) | 95.2 (83.0) |
| Redundancy | 6.6 (6.3) |
| Wilson $B$ (Å$^2$) [direction] | 111 [a*] 71 [b*] 42 [c*] |
| **Refinement** | |
| Copies/a.s.u. | 10 |
| Resolution (Å) | 2.24 Å |
| No. reflections | 63,574 |
| $R_{work}$ / $R_{free}$ (%) | 24.0 / 28.8 |
| No. atoms | 16,195 |
| Protein | 15,574 |
| Ligand | 290 |
| Water | 331 |
| $B$ factors (Å$^2$) | 62.7 |
| Protein (Å$^2$) | 62.9 |
| Ligand (Å$^2$) | 69.7 |
| Water (Å$^2$) | 47.9 |
| R.m.s.d. | |
| Bond lengths (Å) | 0.003 |
| Bond angles (°) | 0.51 |
| Ramachandran (favored/allowed/ outlier) (%) | 98.2/1.8/0 |
| Clashscore | 4.8 |
| Rotamer outliers (%) | 1.6 |

The dataset was collected from a single crystal. Values in parentheses are for the highest-resolution shell.
*a.s.u.* asymmetric unit, *R.m.s.d.* root mean square deviations.

is compatible with compound binding. Following near complete methylation, we complexed methylated UCHL1 with GK13S (Supplementary Fig. 8a) and identified a crystallization condition from which the structure of UCHL1 in complex with GK13S was solved to 2.24 Å resolution after anisotropic scaling (Table 1). The asymmetric unit contained ten copies with a well-defined density of the ligand (in eight copies for warhead, pyrrolidine, central amide, imidazole and phenyl ring and in two copies up to the imidazole; the peripheral amide and the alkyl chain do not adopt a defined position in all copies)

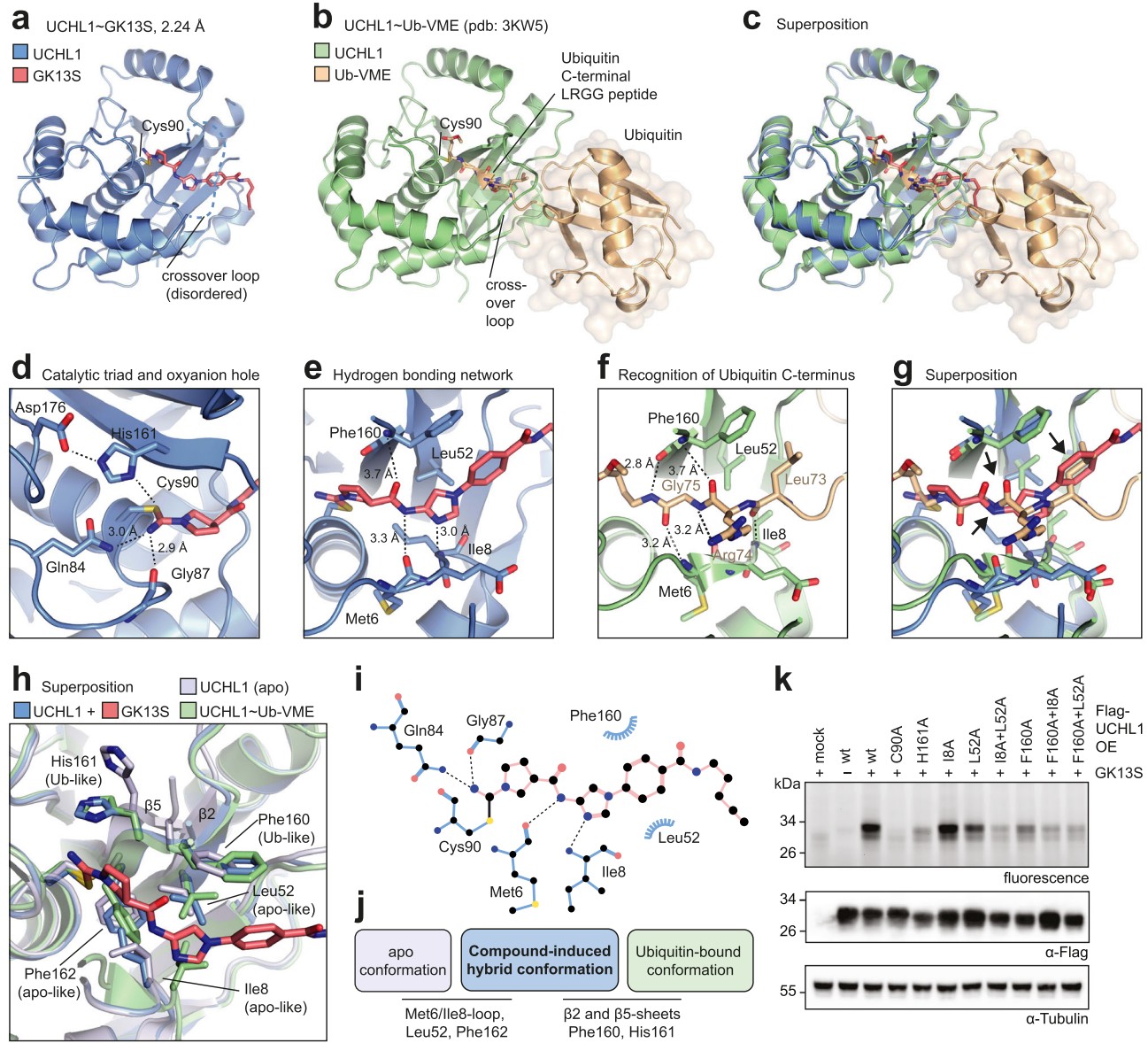

**Fig. 5 | A compound-induced hybrid conformation underlies GK13S-mediated inhibition of UCHL1. a** Structure of UCHL1 (blue) in complex with GK13S (red). The covalently bound active site cysteine 90 and the disordered crossover loop are indicated. **b** Structure of UCHL1 (green) in complex with Ub-VME (gold) (pdb: 3KW5). The Ubiquitin C-terminal residues 73–76 are also shown as sticks. VME vinyl methyl ester. **c** Superposition of a and b, highlighting that the compound binds into the cleft, which is guiding the Ubiquitin C-terminus to the active site. **d** Close-up view on a, highlighting the oxyanion hole (Gln84 and Gly87) bound by the iso-thiourea nitrogen and the aligned catalytic triad residues Cys90, His161, and Asp176. Hydrogen bonds are shown as dotted lines, and their length is given. **e** Close-up view on a, highlighting residues of UCHL1 interacting with GK13S. **f** Close-up view on **b**, highlighting key interactions of the Ubiquitin C-terminus with the binding cleft of UCHL1. **g** Superposition of e and f. Ubiquitin residues mimicked by GK13S are indicated with black arrows (a hydrogen bond acceptor, a hydrogen bond donor and the hydrophobic Leu73 side chain). **h** Superposition of UCHL1-GK13S with apo UCHL1 (pdb: 2ETL) and UCHL1-Ub-VME (pdb: 3KW5). Residues in the UCHL1-GK13S structure are labeled as either apo-like or Ub-like, depending on whether they resemble the orientation observed in the apo or the Ub-VME-bound structures, respectively. **i** 2D representation of the ligand binding pocket observed in the UCHL1-GK13S structure. **j** Schematic overview of structural changes underlying the GK13S-induced hybrid conformation. **k** Validation of the binding site in cells. Indicated Flag-UCHL1 constructs with mutations in the GK13S binding site were overexpressed in HEK293 cells. Cells were treated with a compound where indicated, and GK13S-bound proteins were visualized by in-gel fluorescence. Uncropped versions of gels and blots are shown in the supplementary information.

(Supplementary Fig. 8b–d). Superposition of all copies revealed identical positioning of the ligand (Supplementary Fig. 9a–c).

GK13S is bound covalently to the catalytic cysteine Cys90 through an isothiourea with its cyanamide warhead and occupies a shallow cleft that, during catalysis, guides the Ubiquitin C-terminal LRGG peptide to the active site (Fig. 5a–c)[13]. The isothiourea is stabilized by the oxyanion hole formed by Gln84 and Gly87, with the residues of the catalytic triad in hydrogen bonding distance (Fig. 5d). The central amide as well as the imidazole ring of GK13S engage UCHL1 on both sides of the cleft

with hydrogen bonds to the backbone of Phe160, Met6 and Ile8, which also coordinate the backbone of the Ubiquitin C-terminal peptide (Fig. 5e–g). Superposition revealed that GK13S features hydrogen bonding acceptors and donors of similar geometry as Ubiquitin and that the phenyl ring of GK13S mimics the hydrophobic side chain of Leu73 of Ubiquitin which is recognized by Phe160 (see black arrows in Fig. 5g).

Superposition of the structure of compound-bound UCHL1 with apo (Supplementary Fig. 9d) and Ubiquitin-bound (Fig. 5b, c)

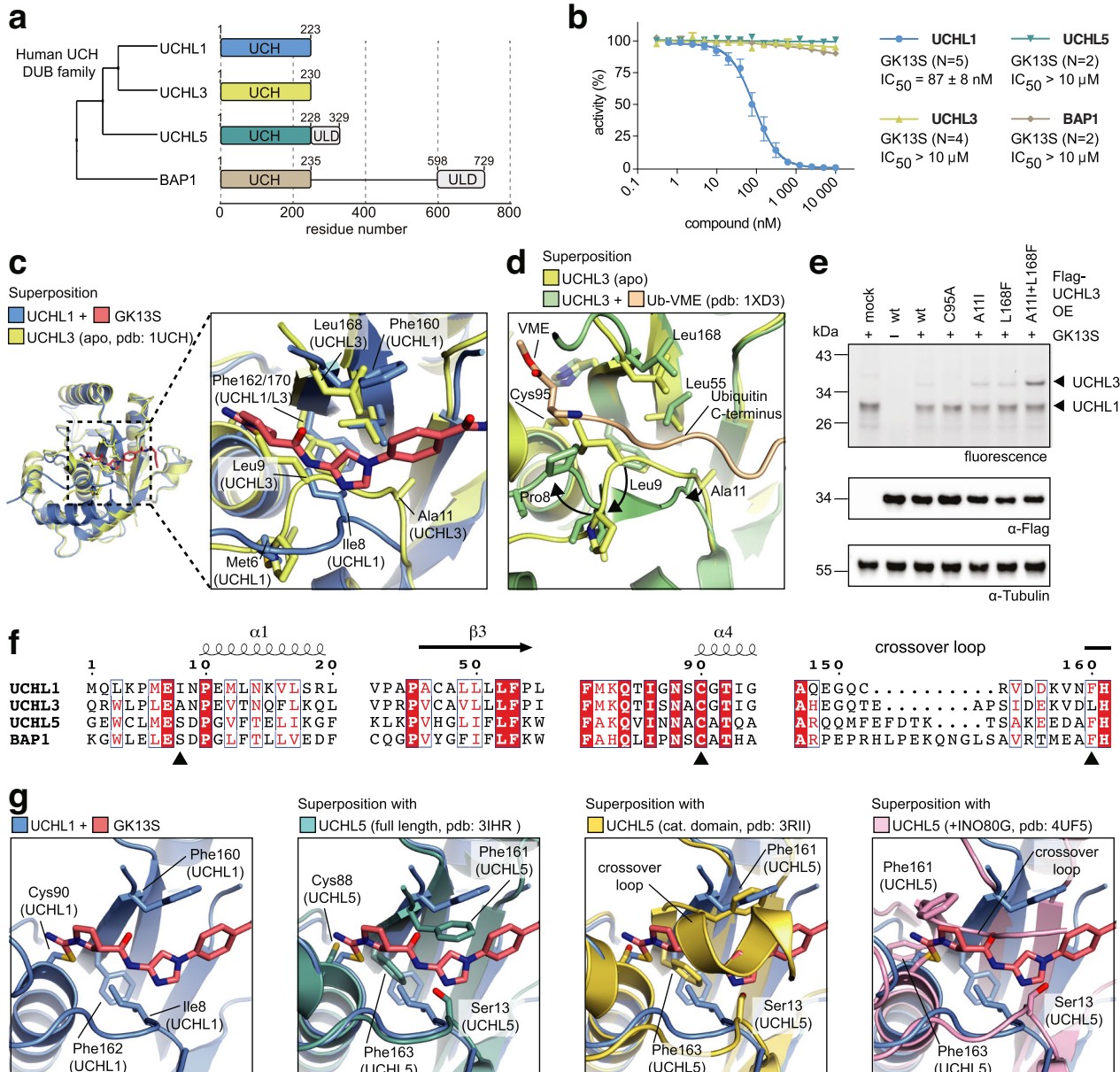

**Fig. 6 | Structural basis for specific inhibition of UCHL1. a** Average distance alignment and schematic representation of domain architecture of human UCH family deubiquitinases. Boundaries of bacterially expressed UCH catalytic domain constructs are given. ULD, UCH37-like domain. **b** Ubiquitin rhodamine cleavage assay of indicated recombinant UCH DUBs, preincubated with GK13S dilutions for 1 h. Data points are shown as mean ± standard error from $N = 2–5$ independent experiments, as indicated in the figure. Source data are provided as a Source Data file. **c** Superposition of UCHL1-GK13S with apo UCHL1 (pdb: 2ETL) and apo UCHL3 (pdb: 1UCH) and close-up view of the binding pocket. Labeled residues are shown as sticks. **d** Superposition of the equivalent to the compound binding pocket in apo UCHL3 and UCHL3-Ub-VME (pdb: 1XD3). The Ubiquitin C-terminus is shown in cartoon representation. Residues undergoing a conformational change upon Ubiquitin binding are indicated with arrows. **e** Indicated Flag-UCHL3 constructs with mutations introducing a GK13S binding site were overexpressed in HEK293 cells. Cells were treated with a compound where indicated, and GK13S-bound proteins were visualized by in-gel fluorescence. Uncropped versions of gels and blots are shown in the supplementary information. **f** Sequence alignment of human Ubiquitin C-terminal Hydrolase (UCH) family members. Secondary structure assignments are based on the UCHL1-GK13S structure. Black arrows indicate residues mutated in **e**. For a full sequence alignment, see Supplementary Fig. 10a. **g** Close-up view of the binding pocket of UCHL1-GK13S (left panel) and super-position with full-length UCHL5 (pdb: 3IHR), the catalytic domain of UCHL5 (pdb: 3RII) and UCHL5 in co-complex with an inhibitory fragment of INO80G (pdb: 4UF5) (from left to right).

structures[13,14] showed overall high similarity, yet closer inspection of the compound binding site revealed characteristic differences (Fig. 5h). The side chains of some residues, in particular of Phe162, Ile8, and Leu52, in the compound-bound structure displayed the same arrangement as observed in apo UCHL1, creating a narrow hydrophobic pocket which surrounds the pyrrolidine of GK13S and accounts for the observed preference for the *S*-configured stereoisomer. Conversely, the side chains of His161 and Phe160 and the secondary structure elements β5 and β2 showed a conformation observed only in

the Ubiquitin-bound state. This allows recognition of the phenyl ring in GK13S by the side chain of Phe160 and hydrogen bonding through its backbone amide. These interactions also enable GK13S to trigger an allosteric relay which was previously[13] described for the coupling of Ubiquitin engagement and catalysis (Supplementary Fig. 9c). Collectively, the structure reveals the striking plasticity of UCHL1, which is locked in a GK13S-induced hybrid conformation comprised of apo and Ubiquitin-bound states (Fig. 5i, j). This hybrid conformation is essential as Ile8 in the Ubiquitin-bound conformation would sterically clash with

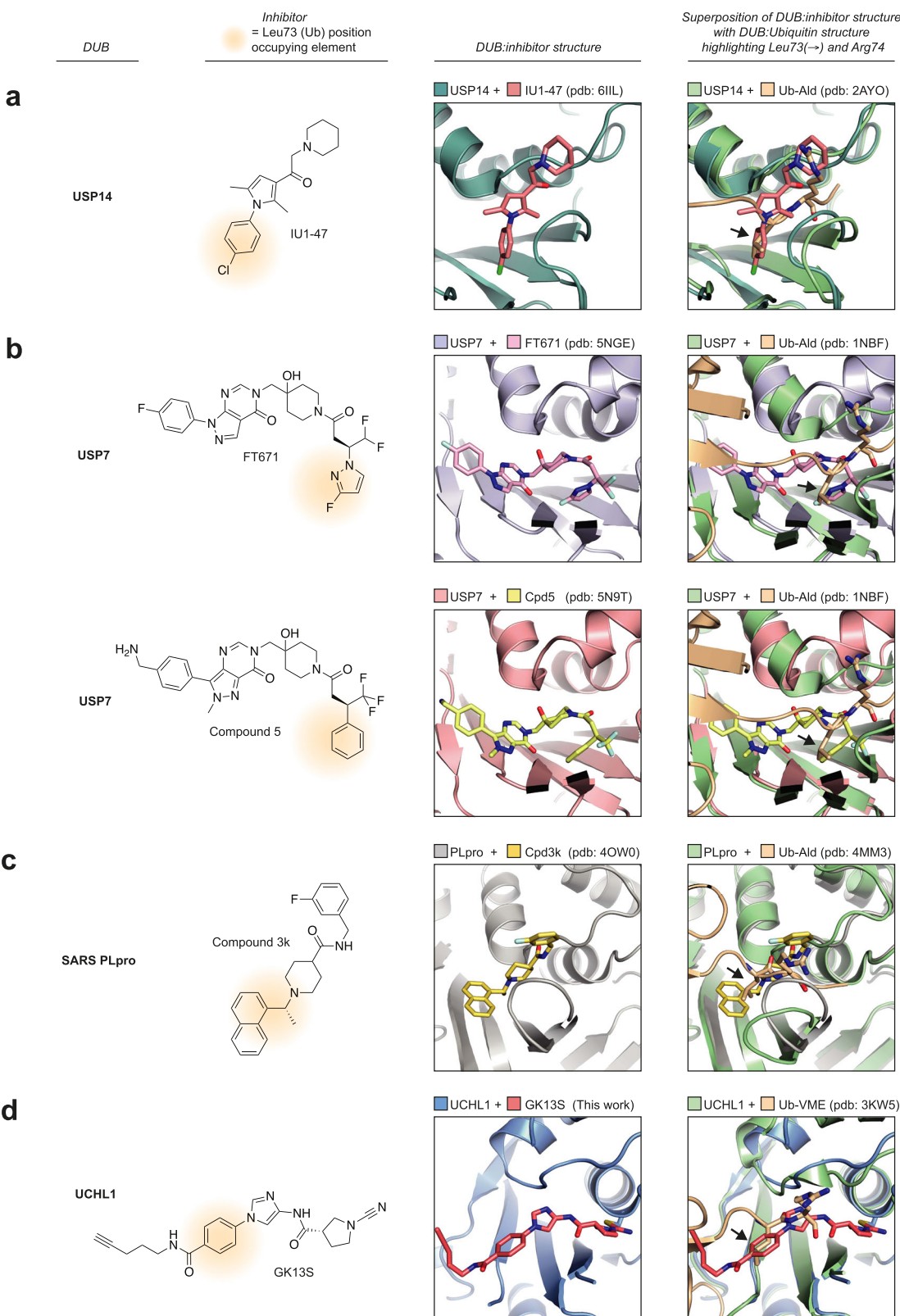

**Fig. 7 | A potential hotspot for DUB ligandability across families. a–d** Selection of structurally characterized DUB inhibitors of USP14[50] (**a**), USP7[44,51] (**b**), SARS PLpro[52] (**c**), and UCHL1 (**d**). Shown are the structures of the compounds, a close-up view of their binding pockets from co-crystal structures, and overlays of these co-crystal structures with Ubiquitin-bound structures of the respective DUBs. The Leu73 position is indicated with black arrows. The chemical moieties of the inhibitors that occupy the position of Leu73 of Ubiquitin are highlighted with a beige background.

the imidazole ring of GK13S, yet Phe160 in the apo conformation would be unable to perform π-π-stacking with the phenyl ring of the compound. We validated the importance of these residues through overexpression of mutated forms of UCHL1 in HEK293 cells and subsequent activity-based protein profiling with GK13S. This led to drastic decreases in fluorescence, indicating reduced compound binding to mutated UCHL1 proteins (Fig. 5k). Of note, the binding site of GK13S to UCHL1 is distinct from that of a peptide fluoromethyl ketone[48], which was soaked into UCHL1 apo crystals and engages hydrophobic residues in neighboring copies within the crystal lattice, but not in the Ubiquitin binding cleft near the modified cysteine (Supplementary Fig. 9e–g).

### Structural basis for specific inhibition of UCHL1

Our structure revealed that GK13S mimics the commonly recognized Ubiquitin C-terminal peptide through hydrogen bonding and hydrophobic interactions, prompting the question of why the probe is specific for UCHL1 in the complex proteome (Fig. 2e). The UCH family of DUBs comprises four members, including UCHL3 as the closest homolog of UCHL1, as well as proteasome associated UCHL5 and the tumor suppressor BAP1 with the latter two featuring extensions C-terminal to the catalytic domain with regulatory functions (Fig. 6a)[2,3]. Assays with purified catalytic domains of these proteins confirmed the exquisite specificity of GK13S for UCHL1 over the other UCH family members (Fig. 6b and Supplementary Fig. 9h).

Structural superposition of compound-bound UCHL1 with apo UCHL3[49] revealed that UCHL3 adopts a different conformation in its N-terminal residues (Fig. 6c), and also a distinct conformational transition occurs upon Ubiquitin binding (Fig. 6d). In particular, Leu9 of UCHL3 (corresponding to Met6 in UCHL1) sterically precludes binding of GK13S to its apo conformation, and the lack of a large hydrophobic side chain of Ala11 of UCHL3 (corresponding to Ile8 in UCHL1) prevents the formation of a hydrophobic pocket for the pyrrolidine ring. Moreover, Leu166 in UCHL3 (corresponding to Phe160 of UCHL1) would form a different environment for the phenyl ring of GK13S. This analysis suggested that the aromatic side chain of Phe160 and the UCHL1-specific apo conformation triggered by Ile8 establish the basis for specificity of GK13S to UCHL1 (Supplementary Fig. 9i, j). To test this hypothesis, we overexpressed UCHL3 with point mutations in these residues exchanged to the corresponding UCHL1 amino acids in HEK293 cells and assessed their probe binding (Fig. 6e, f). While A11I and L168F alone showed weak binding to GK13S, the double-mutant of UCHL3 showed binding to GK13S similar to that of endogenous UCHL1. Consistently, recombinantly purified UCHL3 with the double-mutation could be inhibited by GK13S, albeit with lower potency than wild-type UCHL1, suggesting that the UCHL1-specific, inhibition-competent conformation can at least partially be triggered by these mutations in UCHL3 (Supplementary Fig. 10a–c). In contrast, all UCHL3 variants were inhibited by the non-specific alkylating agent iodoacetamide with similar potency, demonstrating that the binding of GK13S to UCH enzymes is not primarily driven by the chemical reactivity of the catalytic cysteine (Supplementary Fig. 10d).

The activity of UCHL5 is allosterically regulated by conformational changes upon engagement of the proteasome as well as regulatory complexes[2]. Both full-length and catalytic domain structures of UCHL5 in their apo states as well as a complex with INO80G showed an occupied cleft incompatible with GK13S binding. Moreover, the substitution of Ile8 in UCHL1 for a smaller and polar serine (Ser13 in UCHL5 or Ser10 in BAP1) likely prevents hydrophobic coordination of the pyrrolidine as the space is occupied by phenylalanine (Fig. 6g). These differences can rationalize why GK13S does not bind to UCHL5 and BAP1.

Taken together, our data firmly establish that (i) a UCHL1-specific apo conformation, (ii) subtle differences in the binding pocket on both sides of the catalytic cleft within the UCH family of DUBs as well as (iii)

conformational plasticity underly the specific inhibition of UCHL1 by GK13S. They further demonstrate that GK13S is specifically recognized by UCHL1, and that this specificity is not the result of increased reactivity or abundance of UCHL1.

## Discussion

Signaling through ubiquitination requires tight control of deubiquitinases as high constitutive DUB activity may preclude the buildup of specific Ubiquitin modifications. Multiple layers of regulation controlling DUB activity have been described modulating DUB abundance, localization, and catalytic activity[1]. Conformational plasticity, i.e., active and inactive conformations, can structurally enable such regulation and has been reported for multiple deubiquitinases, including UCHL1. While its apo conformation displays an unaligned catalytic triad and a partially occupied binding cleft[14], engagement of the globular part of Ubiquitin facilitates a conformational change which aligns the triad and widens the cleft making it competent for guiding the Ubiquitin C-terminal LRGG peptide to the catalytic cysteine[13]. While for its homolog UCHL5 activating as well as inhibitory regulatory proteins have been described[2], it is unclear whether similar proteins exist for UCHL1, which could allosterically control its conformational state.

We here show that this conformational plasticity underlies the specific inhibition of UCHL1 by the 3-carboxy-N-cyanopyrrolidine probe GK13S, which locks the enzyme in a hybrid conformation of apo and Ubiquitin-bound states. GK13S mimics the C-terminal LRGG peptide of Ubiquitin through key hydrogen bonding and hydrophobic interactions yet achieves specificity through positioning its pyrrolidine in a pocket unique to the apo conformation of UCHL1. Structural information for specific inhibitors with cellular potency is available only for very few other DUBs[22,44], including USP14[50], USP7[44,51], and the coronavirus protease PLpro[52]. It is noteworthy that in these cases, despite the entirely different DUB fold inhibitors are engaged where the Leu73-Arg74 dipeptide of Ubiquitin is recognized, suggesting a general hotspot for DUB ligandability across families (Fig. 7a–d).

Whereas other DUB inhibitors stabilize either the active or the inactive conformation[22,44], the structure of UCHL1 in complex with GK13S highlights that also new (hybrid) compound-induced conformations can be exploited by inhibitors. This principle makes it likely that specific inhibitors will be found for other DUBs which, while sharing their substrate Ubiquitin, may differ in their conformational flexibilities. The structure rationalizes the need for the warhead and the chemical characteristics of its specificity element and explains the observed stereo preference. Moreover, it will form the basis for rational efforts to improve potency and selectivity in further generations of UCHL1 inhibitors.

In addition to UCHL1 as its main protein target, GK13S also binds to PARK7 and the PARK7 family member C21orf33. These targets, as well as the aldehyde dehydrogenases, ISOC1 and NIT2 observed for the control probe GK16S, are in agreement with previous target ID experiments on related 3-carboxy-N-cyanopyrrolidines[27–29]. Non-UCHL1 proteins are also bound by the minimal probe GK16S, which lacks the specificity element, features an IC$_{50}$ for UCHL1 of >100 μM, leads to a much-reduced enrichment in pulldown assays, and, in contrast to GK13S does not inhibit cellular UCHL1. Consistent with the covalent mode of action, we find a reduced specificity window when comparing in vitro potencies to cellular potencies. The large difference in the potencies of GK13S and GK16S both in vitro and in cells allowed for the formation of a chemogenomic pair of probes for the specific investigation of cellular UCHL1 (Fig. 2e, right panel, Fig. 3e, and graphical summary in Supplementary Fig. 10e).

Our findings also suggest that these other proteins are predominantly engaged in a reactivity-driven manner due to the electrophilic nature of the cyanamide. We contrasted the reaction mechanisms of endogenous substrates of PARK7 and of the glutamine amidotransferase C21orf33 to their irreversible reactions with

GK16S (Supplementary Fig. 11), revealing that 1,3-carboxycyanopyrrolidines and glutamine share a similar size and connectivity between their electrophilic center and a carbonyl moiety. Future studies comparing cyanamides of different geometries will need to clarify their target scope, substrate recognition and cellular occupancies of these and other targets. The fact that 3-amino-*N*-cyanopyrrolidines have recently been described as inhibitors for USP30[53] and JOSD1[54] makes it likely that an expanded set of chemogenomic probes could be assembled to specifically investigate the functions of these DUBs despite the cross-reactivity of some of the scaffolds.

Our data establish that both GK13S and GK16S do not impair cell growth in HEK293 and U-87 MG cells under conditions that lead to complete inhibition of UCHL1 by GK13S. This demonstrates that the high toxicity of the 2-carboxy-*N*-cyanopyrrolidine CG173 as well as related compounds is unrelated to UCHL1. CG173 binds twice to UCHL1 in vitro, consistent with modification of both catalytic Cys90 and hyperreactive Cys152[18]. Counterintuitively, CG173 shows weak and decreasing inhibition of cellular UCHL1 with longer incubation times. Whether this behavior and the toxicity are a general feature of the substituted 2-carboxy-*N*-cyanopyrrolidine scaffold[24,27,28] remains to be investigated from larger compound libraries.

Despite intense research efforts, mechanistic links that reconcile the various catalytic activities of UCHL1 with regulated cellular phenotypes are scarce[3]. We concur with others who have called the use of compound LDN-57444 as a specific and effective inhibitor of cellular UCHL1 into question[27]. Genetic models have linked UCHL1 to neurodegeneration[4] and reported reduced levels of monoubiquitin in brain tissue of a UCHL1 mutant mouse[5,6], which was previously studied with overexpressed mutant UCHL1 in a monkey cell line[42]. We reconstituted the effect in a human glioblastoma cell line through controlled small-molecule-mediated inhibition of endogenous UCHL1. This not only validated our chemogenomic pair of probes for the investigation of UCHL1, but also established U-87 MG cells as a suitable model system for UCHL1-dependent phenotypes, such as the observed disturbance in ubiquitin homeostasis.

As GK13S both binds the catalytic cysteine and blocks the engagement of Ubiquitin, it will not only abrogate the hydrolase activity but also other activities. We can, therefore, not conclude whether the reduced levels of monoubiquitin come e.g., from reduced hydrolytic processing of substrates or from reduced binding / buffering of free Ubiquitin. It is tempting to speculate that the large amounts of UCHL1, through regulation of the concentration of available free Ubiquitin, may act as a molecular rheostat which would determine global ubiquitination activities. These mechanisms and the link to neurodegeneration remain to be investigated.

As both probes are equipped with a bioorthogonal handle, their target spectrum in additional cell lines can readily be evaluated as a control experiment. In addition, their covalent nature will facilitate Ubiquitin-probe-based UCHL1 target engagement experiments to ensure that observed effects track with UCHL1 inhibition. We are thus convinced that the here described set of nontoxic, chemogenomic probes for UCHL1 will not only enable the structure-guided improvement of UCHL1 inhibitors but will also be directly and broadly applicable in the dissection of its cellular functions. Moreover, our structure suggests a general hotspot of ligandability within DUBs of different families and that specificity of inhibitors for other DUBs may be achieved through the exploitation of conformational flexibility underlying the endogenous regulation of DUB activity.

## Methods

### Chemical synthesis

See the supplementary information for compound synthesis and characterization data.

### Cloning and constructs

Human UCHL1 (UniProt: P09936, residues: 1-223), UCHL3 (UniProt: P15374, residues: 1-230), UCHL5 (UniProt: Q9Y5K5, residues: 1-228), BAP1 (UniProt: Q92560, residues: 1-235), and PARK7 (UniProt: Q99497, residues: 1-189) sequences were cloned from cDNA templates for bacterial expression into the pOPIN-B vector (for UCHL1 and PARK7, N-terminal His$_6$-3C-tag, an additional GS linker was used for UCHL1) or into the pOPIN-K vector (for UCHL3, UCHL5, and BAP1, N-terminal His$_6$-GST-3C tag) and into the pOPIN-E vector for mammalian cell transfection (with an N-terminal Flag-GS tag and no C-terminal tag) using the In-Fusion HD Cloning Kit (Takara Clonetech). Site-directed mutagenesis was carried out by splicing-by-overlap extension PCR using Phusion Polymerase (New England BioLabs).

### Protein expression and purification

For bacterial expression of proteins, Rosetta2(DE3) pLacI cells were transformed with the respective vector. Overnight cultures were diluted 1:100 into 2xTY medium, which was supplemented with appropriate antibiotics, and cultures were grown shaking at 37 °C. When an $A_{600}$ of 0.8 was reached, cultures were cooled to 18 °C, isopropyl-1-thio-β-D-galactopyranoside (IPTG) was added to a final concentration of 0.5 mM and cultures were grown overnight. Cells were harvested by centrifugation and stored at −80 °C. The pellets were thawed, resuspended in lysis buffer (50 mM H$_2$NaPO$_4$, 300 mM NaCl, 20 mM imidazole, pH 8.0, supplemented with lysozyme and DNAse, and with Complete protease inhibitors for UCHL5 and BAP1) and lysed by sonication on ice for 5 min. The lysates were cleared by centrifugation at 22,000×g for 30 min at 4 °C and sterile filtered. The clear lysate was then passed through a 5 mL HisTrap column (GE Healthcare), preequilibrated with buffer A (50 mM H$_2$NaPO$_4$, 300 mM NaCl, 20 mM imidazole, pH 8.0), using an ÄKTA Pure System (GE Healthcare). The protein was then eluted into buffer B (50 mM H$_2$NaPO$_4$, 300 mM NaCl, 500 mM imidazole, pH 8.0). Protein-containing fractions were pooled and concentrated. For UCHL1 and PARK7, GST-3C protease was added and the sample was dialyzed into buffer C (20 mM Tris pH 8.0, 100 mM NaCl, 4 mM DTT) overnight. These proteins were further purified by size exclusion chromatography using a HiLoad 16/600 Superdex 75 pg column (GE Healthcare) with buffer C. PARK7 was dialyzed subsequently into buffer D (20 mM KH$_2$PO$_4$ pH 7.0 and 5 mM DTT). For UCHL3, UCHL5, and BAP1, His$_6$-3C protease was added and the sample was dialyzed into a lysis buffer. Dialyzed samples were passed through a preequilibrated HisTrap column and the eluate was diluted into a low salt buffer (25 mM Tris pH 8.5, 50 mM NaCl, 4 mM DTT). These proteins were further purified by anion exchange chromatography on a ResQ column (GE Healthcare) by elution into a high salt buffer (25 mM Tris pH 8.5, 500 mM NaCl, 4 mM DTT) over 20 column volumes. Fractions containing pure protein were pooled, concentrated, and buffer exchanged into buffer C + 5% glycerol and protein concentrations were measured by UV absorbance on a Nanodrop 2000.

### Lysine methylation of UCHL1

Methylation of primary amines in UCHL1 was achieved by adding 600 μL of freshly prepared dimethylamine-borane complex (1 M) and 1.2 mL formaldehyde solution (1 M) to 30 mL of UCHL1 (1 mg/mL) in buffer E (50 mM Hepes pH 7.5, 250 mM NaCl). After incubation for 2 h at 4 °C, 600 μL of dimethylamine-borane complex (1 M) and 1.2 mL formaldehyde solution (1 M) were added and the incubation continued for an additional 2 h. Then 300 μL of dimethylamine-borane complex (1 M) was added and the reaction was incubated overnight at 4 °C. The next day precipitated protein was removed by centrifugation. After concentrating to a final volume of 1.5 mL, the protein was purified by size exclusion chromatography into buffer F (50 mM Tris pH 7.5, 200 mM NaCl). The

protein was concentrated at 17 mg/mL and directly used for crystallization experiments.

## Co-crystallization

Crystallization was carried out in 96-well sitting-drop vapor diffusion plates in MRC format (Molecular Dimensions) at 18 °C and set up using a mosquito HTS robot (TTP Labtech). Typical drop ratios of 200 nL + 200 nL and 400 nL + 400 nL (protein solution + reservoir solution) were used for coarse screening and fine screening, respectively. For co-crystallization experiments, methylated UCHL1 (meUCHL1, 17 mg/mL) was preincubated with 1.2 equivalents of GK13S. Covalent adduct formation was confirmed by intact protein mass spectrometry. After buffer exchange into buffer G (20 mM Tris pH 8.0, 100 mM NaCl), meUCHL1-GK13S was concentrated at 30 mg/mL and crystallized in 2.3 M ammonium sulfate, 110 mM $K_3PO_4$, and 90 mM $K_2HPO_4$ as hexagonal prisms ($120 \times 45 \times 45 \, \mu m^3$). Cryoprotection was achieved by placing the crystal for a few seconds into 3 M sodium malonate pH 7.0 with 1 mM GK13S, followed by immediate vitrification in liquid nitrogen.

## Data collection, structure solution and refinement

Diffraction data were collected at 100 K at the Swiss Light Source (SLS, Villigen-PSI, Switzerland) on beamline PXII. The dataset leading to the structure of meUCHL1-GK13S was integrated using Dials[55] and anisotropically scaled using the STARANISO web server[56]. The structure was solved by molecular replacement using MR Phaser[57] and the apoprotein as a search model (pdb 2ETL). Model building using Coot[58] and refinement with Phenix.Refine[59] yielded the final structure. Data collection, anisotropy correction and refinement statistics are given in Table 1. Data have been deposited with the protein data bank under accession code 7ZM0.

## Ub-Rhodamine assay

Reactions were performed in black 384 well low volume non-binding surface plates (Greiner 784900) in a final volume of 20 μL. DUBs were diluted in reaction buffer H (20 mM Hepes pH 8.0, 50 mM NaCl, 0.05 mg/ml BSA) to a 4x stock (final concentrations: UCHL1: 0.06 nM; UCHL3: 0.25–0.5 pM; UCHL5: 0.01 nM; BAP1: 0.01 nM). DUBs were mixed in a 1:1 ratio with 4x compound dissolved in reaction buffer (final DMSO concentration: 0.1–1%). To each well was added 10 μL of DUB-compound solution in triplicates, followed by 1 h incubation time. Reactions were initiated by the addition of 10 μL Ub-Rhodamine 110 (Biomol, final concentration: 50 nM, diluted into reaction buffer supplemented with 5 mM DTT) and fluorescence (excitation = 492 nm, emission = 525) was read on a Tecan Spark plate reader with Tecan SparkControl software for 1 h in 1.5 min intervals at room temperature. Biochemical $IC_{50}$ values were calculated using GraphPad Prism. The experiment with iodoacetamide (Supplementary Fig. 10d) was carried out with a final concentration of 5 mM DTT in buffer H.

## $k_{obs}$/[I] kinetic assay

Reactions were performed in 384 well plates as above. Compound (5 μL of a 4x stock in reaction buffer H supplemented with 2.5 mM TCEP; varying final concentrations ranging from 78 nM to 200 μM) and Ub-Rhodamine 110 (5 μL of a 4x stock in buffer H supplemented with 2.5 mM TCEP; final concentration: 50 nM) were mixed in a 1:1 ratio. To each well was added 10 μL of UCHL1 (2x stock in buffer H supplemented with 2.5 mM TCEP, final concentration: 0.06 nM) and fluorescence was recorded as described above. The kinetic constant $k_{obs}$ was obtained from fitting the curve with a one-phase association function to

$$Y(t) = Y_0 + (A - Y_0) \cdot (1 - e^{-k_{obs} \cdot t})$$

wherein $Y(t)$ denotes the fluorescence change over time $t$, starting at the initial fluorescence $Y_0$ and going up to a plateau $A$. The observed rate constant $k_{obs}$ was plotted over the inhibitor concentration. Linear regression of the corresponding curve resulted in $k_{obs}$/[I]-values as slope, which enabled comparison of covalent inhibitor potencies.

## Intact protein mass spectrometry

The recombinant protein was diluted to a final concentration of 3 or 0.8 μM in buffer I (20 mM Hepes pH 8.0, 50 mM NaCl) and treated with DMSO/compound to result in a final concentration of 10 or 1 μM, respectively. Followed by incubation of 1 h at room temperature, the samples were either run through a MassPrep Online Desalting 2.1 mm × 10 mm cartridge (Waters, flow rate 0.5 mL/min, runtime 7 min, column temperature 30 °C) or an AdvanceBio DesaltingRP 2.1 mm × 12.5 mm cartridge (Agilent, flow rate 0.4 mL/min, runtime 6 min, column temperature 32 °C) with solvents A = HPLC-grade $H_2O$ + 0.1% TFA or formic acid and solvent B = HPLC-grade acetonitrile + 0.1% TFA or formic acid as mobile phases, respectively. A gradient from 20–90% solvent B (MassPrep Online Desalting cartridge) or 5–95% solvent B (AdvancedBio DesaltingRP) was programmed. The samples were either analyzed on a Velos Pro Dual-Pressure Linear Ion Trap mass spectrometer (ThermoFisher, with Xcalibur software), equipped with an electrospray ion source in positive mode (capillary voltage 5 kV, desolvation gas flow 40 L/min, temperature 275 °C) or on an Agilent 1260 II Infinity system (Agilent, with Openlab software), equipped with an electrospray ion source in positive mode (capillary voltage 4 kV, desolvation gas flow 80 L/min, temperature 350 °C). Spectra were deconvoluted with ProMass (Enovatia).

## Thermal shift assay

Reactions were performed in white 96-well PCR plates (Bio-Rad). UCHL1 was diluted in thermal shift buffer (1x PBS, 5 mM DTT) to a 4x concentration of 4 μM. The protein was then mixed with 4x compound dissolved in thermal shift buffer in a 1:1 ratio. To each well was added 20 μL of DUB-Inhibitor solution, followed by 1 h incubation time. To each well was then added 20 μL 4x SYPRO Orange in thermal shift buffer. After sealing the plates with a transparent film, thermal denaturation (gradient: 20–90 °C; Increment: 0.3 °C, hold for 5 s before read) was performed and monitored by a Bio-Rad Connect cycler.

## Jump-dilution assay

Reactions were performed in similar 384 plates as above in a final volume of 20 μL. UCHL1 was diluted in reaction buffer H (20 mM Hepes pH 8.0, 50 mM NaCl, 5 mM HEPES, 0.1 mg/mL BSA) to a concentration of 6 nM (final concentration in well 0.06 nM). To this was added 625 nM GK13S or DMSO and the mixture was incubated for 2 h. Parts of this solution were diluted at 1:50 at different time points with buffer H. All samples were then diluted 1:2 with 100 nM Ub-Rho (buffer H, final concentration in well 50 nM) and fluorescence was recorded as described above. Halt-life time was obtained by fitting the curve with a one-phase decay function to

$$Y(t) = A + (Y_0 - A) \cdot e^{-k_{obs} \cdot t}$$

wherein Y(t) denotes the fluorescence change over time $t$, starting at the initial fluorescence $Y_0$ and decreasing to a plateau $A$. The half-life time displays the time at which the curve reaches 50% of the plateau.

## Dialysis dilution assay

UCHL1 (20 μM) was either incubated with DMSO or GK13S (40 μM) for 1 h to achieve near complete inhibition, and then diluted 1:5 with either PBS or 5 M of urea. A hole was cut into the lid of the 1.5 mL Eppendorf tubes and a SnakeSkin dialysis tubing membrane (3.5 K MWCO) was fit between the lid and the tube. The tubes were turned upside down, fastened in a floating rack and dialyzed against 1 L of PBS overnight.

Protein inhibition was quantified through LC/MS samples which were taken after indicated times of dialysis.

## Cell culture

Cell lines were obtained from the American Type Culture Collection (ATCC) or the Leibniz Institute DSMZ-German Collection of Microorganisms and Cell Cultures GmbH. All cell lines were cultivated in a humidified incubator at 37 °C and 5% $CO_2$. HEK293, HeLa, MCF-7 and U-87 MG cells were cultivated in Dulbecco's modified Eagle's medium (DMEM) supplemented with 10% fetal bovine serum (FBS) and 2% penicillin-streptomycin. PC-3 cells were cultivated in F-12K Nut Mix media supplemented with 10% FBS and 2% penicillin-streptomycin. Cells were tested negative for mycoplasma contamination.

## Transfection

HEK293 cells ($7 \times 10^5$/well) were seeded in six-well plates and cultivated for 24 h. PEI transfecting reagent (Polysciences) was preincubated for 15 min with the vectors in a 200 μL OPTI-MEM medium. Cells were then transfected with vectors and incubated for 24 h. Following the treatment with either compounds or DMSO in fresh media for an additional 24 h, cells were processed as described below. For knockdown of UCHL1 and PARK7, siRNA (scrambled: siGENOME Non-Targeting siRNA Control Pools, D-001206-13-05; UCHL1 smart pool: siGENOME Human UCHL1 siRNA, M-004309-00-0005; PARK7 smart pool: siGENOME Human PARK7 siRNA, M-005984-00-0005; UCHL3 smart pool: siGENOME Human UCHL3 siRNA, M-006059-02-0005) were obtained from Dharmacon. Cells were seeded as described above. About 2 μL of 10 μM siRNA was diluted in a 100 μL OPTI-MEM medium. Additionally, 6 μL RNAiMAX (Thermo Fisher) were diluted with 100 μL OPTI-MEM medium. Both solutions were combined, incubated for 5 min, and added dropwise to the cells. Twenty-four hours after transfection, cells were treated with the compounds or DMSO in fresh media for 24 h. Cells were subsequently processed as described below.

## Cell growth and viability assay

HEK293 cells ($5 \times 10^3$/well) or U-87 MG cells ($5.5 \times 10^3$/well) were seeded into 96-well dishes. The following day cells were treated with the compounds at varying concentrations and propidium iodide at 20 μg/mL. The plates were immediately transferred to the Incucyte S3. Bright-field and red fluorescent images were automatically captured at 1 or 2 h intervals for 72 h with 10x magnification. The confluency of cells and the number of PI-positive cells was determined using the Incucyte masking program.

## Activity-based protein profiling in the lysate

Cells ($4 \times 10^6$) were seeded in 10 cm dishes and grown to 90% confluency. The medium was aspirated, cells were washed with ice-cold PBS, treated with 600 μL ABP lysis buffer (1% (v/v) IGEPAL, 50 mM Tris, 150 mM NaCl, 5% glycerol, pH 8, cOmplete protease inhibitor cocktail), and incubated on ice for 15 min. The lysed cells were scrapped off the dish, cleared by centrifugation, and the protein concentration was determined via a Bradford assay. The cell lysate was diluted to a protein concentration of 2–4 mg/mL with ABP lysis buffer, split in half, and one-half treated with HA-Ub-VS (1 μM final concentration, 37 °C, 30 min). Each compound was diluted to a 2x concentration in PBS buffer from a 10 mM stock in DMSO (final concentration: 1 μM). Then each compound dilution or a DMSO dilution was mixed with cell lysate and incubated for one hour at room temperature. Afterwards, 1 μL of each click reagent (final concentration of 0.5 mM $CuSO_4 \cdot 5\ H_2O$, 1 mM BTTAA, 4 μM 5/6-TAMRA-Azide-Biotin (Jena Bioscience), 5 mM sodium ascorbate, each from 100x stocks) were added to each sample, followed by an incubation period of one hour at room temperature with protection from light. Samples were resolved by SDS-PAGE using a 4–12% Bis-Tris gel (Invitrogen, NuPAGE) with MES SDS running buffer for 50 min at 180 V. Fluorescence was read out using the Alexa564 ($\lambda_{ex/em}$ 520–545/

577–613 nm, for TAMRA) and Alexa680 ($\lambda_{ex/em}$ 650–675/700–730 nm, for the PageRuler Prestained NIR Protein Ladder (Thermo Scientific)) channels on a ChemiDoc MP Imaging System (Bio-Rad).

## Cellular activity-based protein profiling

Cells ($7 \times 10^5$) were seeded in six-well plates and cultivated for 48 h. Cells were then incubated in fresh DMEM supplemented with compound or DMSO for 24 h. The cells were washed with ice-cold PBS and harvested in 200 μL ABP lysis buffer as described above. The cell lysate was diluted to a protein concentration of 2 mg/mL with ABP lysis buffer. One microliter of each click reagent (see above) were added to each sample, followed by an incubation period of one hour at room temperature. Samples were separated by SDS-PAGE and analyzed for fluorescent protein-compound conjugates as described above.

## Western blotting

Proteins were transferred to a polyvinylidene fluoride (PVDF, only experiments in Fig. 4e and Supplementary Fig. 7e, f) or nitrocellulose (all other experiments) membrane using a Trans-Blot Turbo system (Bio-Rad, 1.3 A, 25 V, 10 min). The membranes were blocked with 5% (m/v) nonfat milk in PBS-T buffer and incubated with indicated primary antibodies (anti-UCHL1, 1:1000, Cell Signaling, D3T2E; anti-PARK7, 1:1000, Cell Signaling, D29E5; anti-Tubulin, 1:4000, Sigma, T6199; anti-Hemagglutinin, 1:1000, BioLegend, 16B12; anti-flag, 1:2000, Sigma, F3165; anti-Ubiquitin, 1:1000, Cell Signaling, P4D1; anti-Ubiquitin, 1:300, Santa Cruz, P4D1, sc-8017; anti-UCHL3, 1:1000, Proteintech, 12384-1-AP) overnight. Then the membranes were incubated with the respective secondary antibody (anti-mouse,1:5000, Sigma, NXA931; anti-rabbit, 1:5000, Sigma, GENA934) coupled to horseradish peroxidase. The chemiluminescent reaction was initiated using a Clarity Western ECL substrate (Bio-Rad) and images were taken on a ChemiDoc MP Imaging System (Bio-Rad).

## Cellular Ub-probe competition

Cells were cultured, treated with a compound, and lysed as described above. The total protein concentration was adjusted to 2 mg/ml by diluting each sample with ABP lysis buffer. A final concentration of 1 μM HA-Ub-VS probe was added, followed by incubation for 30 min at 37 °C. The labeling reaction was quenched by the addition of 4x LDS sample buffer. The samples were separated via SDS-PAGE and further analyzed via western blotting as described above.

## Identification of probe-labeled proteins with mass spectrometry

Cells were cultured, treated with a compound, and lysed as described above. The cell lysate was diluted to a protein concentration of 2 mg/mL with ABP lysis buffer. 1 μL of each click reagent (see above) were added to each sample, followed by an incubation period of one hour at room temperature with protection from light. The sample volume was adjusted to 1000 μL with PBS and 30 μL of a NeutrAvidin (Thermo Fisher) bead slurry (prewashed 3x with PBS) were added to each sample. The samples were incubated for one hour to overnight on a rotator at 15 rpm at 4 °C. Beads were pelleted by centrifugation, the supernatant was discarded, and the beads were washed six times (1x with ½x lysis buffer, followed by five washes with PBS). After removing the washing solution completely, the beads were subjected to reduction with dithiothreitol (1 mM), alkylation with chloroacetamide (5 mM) and on-bead digestion with first LysC (Serva Biotech, 1 h, 37 °C), followed by trypsin (Sigma Aldrich, 1 h, overnight). Tryptic peptides were desalted with C18 StageTips and analyzed by nano-HPLC-MS/MS.

An Ultimate 3000 RSLC nano-HPLC system and a Hybrid-Orbitrap mass spectrometer (Q Exactive Plus) equipped with a nano-spray source (Thermo Fisher Scientific) was used. The protein fragments were enriched on a C18 PepMap 100 column (5 μm, 100 Å, 300 μm ID * 5 mm, Dionex) using 0.1% TFA, at a flow rate of 30 μL/min, for 5 min and separated on a C18 PepMap 100 column (3 μm, 100 Å, 75 μm

ID × 50 cm) using a linear gradient (5–30% ACN/H$_2$O + 0.1% formic acid over 90 min) with a flow rate of 300 nL/min. The nano-HPLC apparatus was coupled online with the mass spectrometer using a standard coated Pico Tip emitter (ID 20 µm, Tip-ID 10 µM, New Objective). Signals in the mass range of m/z 300 to 1650 were acquired at a resolution of 70,000 for the full scan, followed by ten high-energy collision-dissociation (HCD) MS/MS scans of the most intense at least doubly charged ions at a resolution of 17,500. Proteins were relatively quantified by using MaxQuant[60] v.2.0.3.1, including the Andromeda search algorithm and searching the *Homo sapiens* reference proteome of the UniProt database. Briefly, an MS/MS ion search was performed for enzymatic trypsin cleavage, allowing two missed cleavages. Carbamidomethylation was set as a fixed protein modification, and oxidation of methionine and acetylation of the N-terminus were set as variable modifications. The mass accuracy was set to 20 parts per million (ppm) for the first search and to 4.5 ppm for the second search. The false discovery rates for peptide and protein identification were set to 0.01. Only proteins for which at least two peptides were quantified were chosen for further validation. Relative quantification of proteins was performed by using the label-free quantification algorithm implemented in MaxQuant. Statistical data analysis of pulldown samples was performed using Perseus[61] v.1.6.15.0, including proteins which were identified in at least four of the five biological replicates which were used per condition. Label-free quantification (LFQ) intensities were log-transformed (log2); replicate samples were grouped together. Pairwise comparisons of groups were performed separately. Missing values were imputed using small normally distributed values (width 0.3, downshift 1.8) and a two-sided *t*-test (s0 = 5, FDR = 0.001) was performed. Enrichment numbers of different proteins are difficult to compare as some (e.g., UCHL1) were measured in the DMSO control samples (owing e.g. to high abundance in proteome), while for others, the enrichment number is a result of the imputation as the protein was not quantified in the control condition. The three most enriched proteins for GK13S (UCHL1, PARK7, and C21orf33) were observed not only in the shown experiment with biological replicates, but also in two other fully independent experiments.

### Quantification of monoubiquitin levels in U-87 MG cells

U-87 MG cells (5 × 10$^5$/well) were seeded in six-well plates and cultivated for 8 h. Cells were transfected with siRNA as described above. On the next day, cells were cultured in fresh media supplemented with either compounds (final conc. 5 µM) or DMSO, where indicated, for a further 48 h. The media was changed every 24 h to a fresh medium supplemented with compounds (final conc. 5 µM) or DMSO. Cells were washed with ice-cold PBS (1x) and lysed for 15 min at 4 °C in 100–200 µL ABP lysis buffer (50 mM Tris pH 7.5, 150 mM NaCl, 5% (w/v) glycerol, 1% (v/v) IGEPAL, cOmplete protease inhibitor cocktail supplemented with 2 mM EDTA, 10 mM chloroacetamide (CAA). Lysed cells were scrapped off the dish, cleared by centrifugation, and the protein concentration was determined via a Bradford assay. The cell lysate was diluted to a protein concentration of 1.3–2.0 mg/mL with ABP lysis buffer. Proteins were separated via SDS-PAGE and analyzed via western blot as described above. Densitometric quantification of bands was carried out using ImageJ (version 1.53o). Monoubiquitin band intensities were first normalized to α-tubulin or total protein (Fast Green FCF, TCI) and subsequently normalized to the intensities of monoubiquitin bands in the sample treated with DMSO or siScr (where no DMSO was used) which was set to "1".

### Statistics and reproducibility

Statistical significance of monoubiquitin changes was analyzed using a one-sample, two-tailed *t*-test compared to a hypothetical mean of "1" as set for the DMSO or siScr samples using GraphPad Prism.

All observations reported in this manuscript were made in at least two independent experiments, typically with technical triplicates, all

with consistent results. Where possible, values were obtained as averages from multiple independent experiments as stated in the Figure legends.

### Reporting summary

Further information on research design is available in the Nature Research Reporting Summary linked to this article.

## Data availability

Data related to the structure of methylated UCHL1 in complex with GK13S have been deposited with the protein data bank under accession code 7ZM0. Proteomics data have been deposited with ProteomeXchange under accession codes MSV000090044 and MSV000090045. Chemical characterization data, as well as uncropped gels and blots, are provided in the Supplementary Information. All data were available on request from the authors. Protein sequences are available through the uniport database under the following accession codes: UCHL1: P09936; UCHL3: P15374; UCHL5: Q9Y5K5; BAP1: Q92560; and PARK7: Q99497. Source data are provided with this paper.

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

## Acknowledgements

We thank the beamline scientists at the Swiss Light Source (SLS) for support during data collection and Raphael Gasper-Schönenbrücher for support with crystallization and biophysics. We are grateful for access to equipment and scientific advice from Daniel Rauh, Stefan Raunser, and Herbert Waldmann. We thank all staff at the Technical University and the Max Planck Institute for their excellent support. We are grateful to all members of the Gersch lab for discussions, advice, and reagents. This work was funded by Deutsche Forschungsgemeinschaft (DFG, German Research Foundation, Project-ID GE 3110/1-1—Emmy Noether, to M.G.). Work in the Gersch lab is further supported by AstraZeneca, Merck KGaA, Pfizer Inc., and the Max Planck Society as part of the Chemical Genomics Center III (CGCIII-352S to M.G.) and by the Deutsche Forschungsgemeinschaft (DFG, German Research Foundation, Project-ID 424228829—SFB1430, to M.G.). We also acknowledge financial support by Deutsche Forschungsgemeinschaft and Technische Universität Dortmund/TU Dortmund University within the funding program Open Access Costs.

## Author contributions

C.G., M.S., and G.-M.K. synthesized compounds. C.G. performed in vitro characterization and crystallization experiments. M.S. performed protein profiling, cellular target ID, and target validation experiments. R.O.D. performed cell viability measurements. K.G. with M.S. performed cellular characterization experiments. P.J. analyzed proteomics samples. All authors planned experiments and analyzed data. M.G. supervised the project and wrote the manuscript with input from all authors.

## Funding

## Competing interests

The authors declare no competing interests.
