## [Peer Review File · Nature Communications]

Structural basis for specific inhibition of the deubiquitinase UCHL1Reviewer #1 (Remarks to the Author):

The manuscript by Grethe and Schmidt et al reports their discovery of a cyanopyrrolidine based best-in-class small-molecule activity-based probe for selective interrogation of UCHL1 cellular function. The report presents development of a pair of chemogenomic tool compounds with favorable characteristics that will allow their use as cellular probe for UCHL1 related processes. In spite of being one of the most studied DUBs, perhaps one of the earliest DUBs to have been discovered, its biological functions remain mysterious. An abundant protein in neurons, the DUB has been linked to neurodegeneration and variety of cancer conditions. It is also one of those rare DUBs, like its close related cousin, UCHL3, which does not show polyubiquitin processing activity that is typically seen in other DUBs. Its substrates are thought to be small-peptide/nucleophile conjugates of ubiquitin but no specific substrate or a set of substrates have been identified so far. A good small-molecule probe along with a suitable control would be highly beneficial in elucidation of role of UCHL1's enzymatic activity in biological pathways.

Starting out with a select group of nitrile-containing covalent inhibitors reported in patent literature, the authors synthesized corresponding alkyne-tagged activity probes, which they applied to a complex proteome present in HEK293 lysate in order to capture irreversibly bound targets, DUBs and other reactive-cysteine proteins/enzymes, in an unbiased manner. This led to a 3-carboxy cyanopyrrolidine based compound as a potent UCHL1 inhibitor, GK13S, a compound (derived from the parent compound, Cpd158, in the patent literature) structurally similar to other compounds in the same patent source that have been studied earlier by three different groups for their UCHL1-selective properties. As a control, they designed another compound, GK16S, containing the same reactive electrophile but lacking certain key elements of GK13S.

Using quantitative proteomics of covalently pulled down proteins, they are able to identify a few proteins enriched by both probes but UCHL1 as the only non-overlapping, selective target of GK13S. Thus, they argue that this compound pair can be used for selective probing of UCHL1's cellular function. To illustrate this point, they treated a glioblastoma cell line with GK13S or the control and showed a reduction in mono-ubiquitin levels (free ubiquitin pool) only upon treatment by GK13S. The results amount to compound-mediated inhibition phenocopying genetic deletion of UCHL1 in the GAD mouse model.

The authors have provided clear biochemical evidence of the mechanism of action and performed the required characterization studies one would expect of this sort of inhibitors. They have shown that the mode of inhibition is covalent irreversible and is stereoselective in its inhibitory activity.

Finally, the authors have co-crystallized GK13S with UCHL1. The structure shows interactions with the inhibitor at the active site and reveals the inhibited enzyme locked in some sort of a hybrid state between apo and ubiquitin bound positions of certain active site residues. They propose that this ability to induce or select this hybrid state is a contributing factor of UCHL1 selectivity.

The structural data are solid with very reasonable interpretations. This is also true of their other results. The biochemical and cellular studies are rigorous and well controlled. Overall, this is an interesting paper that will be enthusiastically received by a broad audience including those interested in UCHL1 biology.

I ask the authors to consider the following points to improve their manuscript.

UCHL1 Selectivity: Biochemical comparison of inhibition across UCH family
The authors argue that GK13S is 'exquisitely specific' for UCHL1 compared to other UCH members. This appears to be the case from their cellular studies (Figure 1D) as no detectable bands for UCHL3, UCHL5 and Bap1 were observed except for UCHL1.

However, it is likely that these DUBs are not around very much in the cell line under investigation (UCHL1 is kind of an abundant protein that shows up readily in many such experiments). Presenting a comparison of inhibitory potency with purified enzymes will reveal a better sense of biochemical specificity which can be related to crystal structures. Along this line, I would consider performing a similar analysis as in Figure 1D in EL4 cells where there seems to be a higher expression of diverse DUBs including USPs.

I would urge to include the biochemical inhibition assay comparing the UCH members (for Bap1, the catalytic domain will suffice) as part of the main text.

I suspect a part of selectivity could be attributed to differential reactivity of the catalytic cysteine in different UCH DUBs. This needs to be considered.

Free ubiquitin levels in glioblastoma cells

For this analysis (Figure 4F), it would be necessary to include a control where DJ-1 has been knocked down. Since the compounds are likely hitting DJ-1 as well under these conditions, it is important to demonstrate the reduction in free ubiquitin is solely due to UCHL1 inhibition.

Crystallography

The maps don't quite look as well as one expects of a 2.24 Å structure. Is it due to diffraction anisotropy? Also, where did the ordered waters go? I did not see any modeled in the active site area next to the compound. There may be water-mediated interactions worth mentioning.

In Supplementary Figure 6D, please zoom in close to the compound so the density envelope can be better appreciated.

In the same figure, include a new digram showing overlay of key residues (Phe160, His161, etc) in different subunits. This may show protein dynamics in these residues that may explain inhibitor occupancy.

Reviewer #2 (Remarks to the Author):

The manuscript by Malte Gersch and colleagues describes novel small molecule activity based probes and their structure-based specific interactions with UCH-L1, a ubiquitin C-terminal hydrolase belonging to the deubiquitylating enzyme (DUB) family. UCHL1 has been of considerable interest for a long time as a potential target in neurodegeneration, but also cancer metastasis and invasiveness.

This study has been carried out with a very high standard in the quality of biochemical and structure-based experiments, and the characterisation of the lead compounds/probes to target UCH-L1 as well as their molecular basis of specific binding has been carried out in a thorough manner. This work potentially reflects a nice contribution to the ubiquitin research community.

However, before this work can be considered for publication, there are a number of points that should be addressed as outlined below.

Specific points:

1. Figure 1 and subsequent results indicate additional non-DUB targets for these

compounds, such as PARK7 and C21orf33. What is the chemical basis for this potential cross-reactivity? For instance, PARK7 may have a glyoxalase/deglycase activity. The authors should discuss potential hydrolase mechanism of action and cross-reactivity as additional information may help with a further refinement of such probes in the future.

2. This is not the first report describing small molecular probes & inhibitors against UCH-L1. A comparison of GK to previously characterised inhibitors and probes, such as isatin O-acyl oxime compound (LDN-5744) (Liu et al., *Chem Biol.* 2003 Sep;10(9):837-46.) or IMP-1710 (Nattawadee et al., *J. Am. Chem. Soc.* 2020, 142, 28, 12020–12026) would be very insightful, in particular when describing the molecular basis for selectivity.

3. Proteomics data used in the present study (method described in lines 1075-1137) is recommended to be deposited into a public repository (e.g. PRIDE) or similar.

We would like to thank the Reviewers for their highly insightful, detailed, and supportive comments, which we have addressed in full and with additional data. Please find below a point-by-point response to the comments.

We would like to make the reviewers aware of a data exchange which we have carried out during the revision at our own initiative. This exchange did not limit or substantially change the conclusions and is described in the following paragraphs:

We chose to exchange the proteomics dataset to a repeat whose measurement was completed during the reviewing phase. In the previously used dataset, UCHL1 was measured at LFQ intensities of 30-34 in all replicates which is extremely high and close to the upper limit of the dynamic range of the used spectrometer. These numbers make sense given that the probe enriches a very abundant protein. However, the used conditions also led to a large amount of UCHL1 in the DMSO-treated samples, which limited the observed enrichment factor to approx. 4 (on a log₂ scale) which is not in line with the gel-based results. Moreover, the large amount of UCHL1 did seem to limit the enrichment or detection of less abundant targets. For example, the gel-based results shown in Figure 2b clearly show a larger background at 10 μ M, and bands in the 55-70 kDa range, for which however no protein targets were identified in the GK13S sample. We therefore optimized the procedure (sample / beads ratio, higher dilution during binding, more washing, less amount injected into spectrometer) and processed samples with 5 biological replicates at 5 μ M and 1 μ M compounds each, to increase the detection capacity.

Consistent with the previous data and the conclusions presented, UCHL1 is the main protein bound by GK13S at 5 μ M (Figure 2d) with an enrichment factor of 11.5 (equivalent to approx. 3000-fold). The next most enriched proteins are PARK7 and C21orf33 as described before. At this high concentration we now detect a number of other proteins at lower enrichment factors which is consistent with the gel-based experiments. In the 1 μ M samples (Figure 2e), these three proteins are still most highly enriched, but we in addition also detect several aldehyde dehydrogenases and metabolic enzymes (Omega-Amidase NIT2, Isochorismatase domain containing proteins ISOC) which chemically makes sense and is consistent with previous results on related compounds (see our response to reviewer 2 below). Importantly, these still cancel out when the two probes are compared.

One important difference in the dataset is that under the improved conditions we were able to detect UCHL1 also significantly enriched by the control compound GK16S with an enrichment factor of 3 (on a log₂ scale). However, at the same concentration GK13S leads to an enrichment of > 8, so that the main finding of the manuscript still holds true that these compounds can be used as chemogenomic probes (see GK13S-GK16S panel in Figure 2e, showing that UCHL1 in the new dataset still is the main difference between in the bound proteins, consistent with cellular evaluation in Figures 3e and 4e).

We have therefore reworded the manuscript to be more nuanced. While UCHL1 could be detected in the GK16S sample and clearly is bound to a very small yet detectable degree, the cellular inhibition data show that after 24 h GK16S does not bind a substantial fraction of UCHL1 to lead to detectable inhibition in HEK293 cells (in U-87 MG cells, however, a very small yet detectable amount is inhibited, see Figure 4d). Since covalent inhibitors are time-dependent, it makes sense that the large specificity window determined from in vitro assays at 1 h incubation (see Figure 3b, IC₅₀ values of 50 nM vs > 100 μ M for GK13S and GK16S, respectively) is considerably smaller in cells (change in log₂ enrichment values of 8-3 = 5, equating to >32 fold difference). This also explains why the enantiomer GK13R (despite having a 40-fold reduced potency in vitro) is not a robust control in cells where it also leads to complete inhibition of UCHL1 after 24h and an enrichment difference of only 2 (Supplementary Figure 3b). We therefore believe that the here described finding, that the minimal probe GK16S can be used as a suitable control for the cellular investigation of UCHL1, is even more relevant in light of the presented data due to the larger cellular specificity window. Moreover, the presented data are now thoroughly consistent across the gel-based and mass-spectrometry-based analyses.

We are convinced that these changes along with the additions suggested by the reviewers have considerably strengthened and improved this manuscript through the revision, and we hope that it will be deemed suitable for publication in the revised form.

REVIEWER COMMENTS

Reviewer #1 (Remarks to the Author):

The manuscript by Grethe and Schmidt et al reports their discovery of a cyanopyrrolidine based best-in-class small-molecule activity-based probe for selective interrogation of UCHL1 cellular function. The report presents development of a pair of chemogenomic tool compounds with favorable characteristics that will allow their use as cellular probe for UCHL1 related processes. In spite of being one of the most studied DUBs, perhaps one of the earliest DUBs to have been discovered, its biological functions remain mysterious. An abundant protein in neurons, the DUB has been linked to neurodegeneration and variety of cancer conditions. It is also one of those rare DUBs, like its close related cousin, UCHL3, which does not show polyubiquitin processing activity that is typically seen in other DUBs. Its substrates are thought to be small-peptide/nucleophile conjugates of ubiquitin but no specific substrate or a set of substrates have been identified so far. A good small-molecule probe along with a suitable control would be highly beneficial in elucidation of role of UCHL1's enzymatic activity in biological pathways.

Starting out with a select group of nitrile-containing covalent inhibitors reported in patent literature, the authors synthesized corresponding alkyne-tagged activity probes, which they applied to a complex proteome present in HEK293 lysate in order to capture irreversibly bound targets, DUBs and other reactive-cysteine proteins/enzymes, in an unbiased manner. This led to a 3-carboxy cyanopyrrolidine based compound as a potent UCHL1 inhibitor, GK13S, a compound (derived from the parent compound, Cpd158, in the patent literature) structurally similar to other compounds in the same patent source that have been studied earlier by three different groups for their UCHL1-selective properties. As a control, they designed another compound, GK16S, containing the same reactive electrophile but lacking certain key elements of GK13S.

Using quantitative proteomics of covalently pulled down proteins, they are able to identify a few proteins enriched by both probes but UCHL1 as the only non-overlapping, selective target of GK13S. Thus, they argue that this compound pair can be used for selective probing of UCHL1's cellular function. To illustrate this point, they treated a glioblastoma cell line with GK13S or the control and showed a reduction in mono-ubiquitin levels (free ubiquitin pool) only upon treatment by GK13S. The results amount to compound-mediated inhibition phenocopying genetic deletion of UCHL1 in the GAD mouse model.

The authors have provided clear biochemical evidence of the mechanism of action and performed the required characterization studies one would expect of this sort of inhibitors. They have shown that the mode of inhibition is covalent irreversible and is stereoselective in its inhibitory activity.

Finally, the authors have co-crystallized GK13S with UCHL1. The structure shows interactions with the inhibitor at the active site and reveals the inhibited enzyme locked in some sort of a hybrid state between apo and ubiquitin bound positions of certain active site residues. They propose that this ability to induce or select this hybrid state is a contributing factor of UCHL1 selectivity.

The structural data are solid with very reasonable interpretations. This is also true of their other results. The biochemical and cellular studies are rigorous and well controlled. Overall, this is an interesting paper that will be enthusiastically received by a broad audience including those interested in UCHL1 biology.

We are very grateful to the reviewer for the very careful description of our work and for the overly positive judgement.

I ask the authors to consider the following points to improve their manuscript.

UCHL1 Selectivity: Biochemical comparison of inhibition across UCH family

The authors argue that GK13S is 'exquisitely specific' for UCHL1 compared to other UCH members. This appears to be the case from their cellular studies (Figure 1D) as no detectable bands for UCHL3, UCHL5 and Bap1 were observed except for UCHL1. However, it is likely that these DUBs are not around very much in the cell line under investigation (UCHL1 is kind of an abundant protein that shows up readily in many such experiments). Presenting a comparison of inhibitory potency with purified enzymes will reveal a better sense of biochemical specificity which can be related to crystal structures. Along this line, I would consider performing a similar analysis as in Figure 1D in EL4 cells where there seems to be a higher expression of diverse DUBs including USPs.

I would urge to include the biochemical inhibition assay comparing the UCH members (for Bap1, the catalytic domain will suffice) as part of the main text.

I suspect a part of selectivity could be attributed to differential reactivity of the catalytic cysteine in different UCH DUBs. This needs to be considered.

The reviewer raised important points regarding the specificity of the compounds which we have addressed in four different ways along the suggested lines.

1.

We have purified recombinantly expressed catalytic domains of all other human UCH DUBs, and performed *in vitro* inhibition assays with these and Ubiquitin rhodamine as substrate. While GK13S inhibited UCHL1 with an IC_{50} of 87 nM (following a standard pre-incubation of 1 h), none of the other enzymes was inhibited at concentrations up to 10 μ M. Moreover, only UCHL1 was covalently modified by GK13S when incubate with 10 μ M of compound for 1h. See new Figures 6a and b, and new Supplementary Figure 9h.

This is fully in line with the statement of GK13S being exquisitely specific for UCHL1 compared to other UCH members.

2.

The reviewer suggested that the lack of labeling of other UCH enzymes may be because UCHL1 is much more abundant than the others. We hence overexpressed Flag-tagged catalytic domain-containing constructs of all four human UCH DUBs in HEK293 cells and carried out cellular labeling studies. These showed that elevated protein levels did not lead to any appreciable labeling of the other UCH DUBs compared to that of endogenous UCHL1.

This confirms the conclusion that UCHL1 is not labeled as it is the most abundant DUB, but rather due to specific recognition by GK13S. Since cellular levels of the different proteins could not be titrated easily in our transient overexpression system, we have refrained from including these data in the manuscript and provide it here for consideration by the reviewers.

3.

The reviewer suggested to consider a similar *in vitro* analysis as in Figure 1d in EL4 cells, suspecting a higher expression of diverse DUBs including USPs. The experiment in Figure 1d is carried out in HEK293 cell lysate which contains more than 60 active DUBs (including >35 USP DUBs), as identified e.g. by Hewings et al. *Nat Comm* **2018**, 9, 1162 from Ubiquitin probe pull-downs. We would therefore emphasize that HEK293 cells do express a rather diverse collection of active DUBs.

Since we did not have immediate access to the EL4 cell line (which is a mouse cell line, and we have no knowledge of how well GK13S inhibits mouse UCHL1), we conducted a similar *in vitro* analysis as suggested in three other human cell lines: PC-3, HeLa and MCF-7. See new Supplementary Figure 2a-c.

These experiments showed that there is only one Ub-VS competitive band in PC-3 lysate from labeling with GK13S with a molecular weight which is consistent with UCHL1. HeLa and MCF-7 cells are known to have a very low expression of UCHL1 and show no probe-competitive band despite a broad set of active DUBs incl. USPs being present (compare results from Pinto-Fernandez et al. *Front Chem* **2019**, 7, 592 for DUB levels).

4.

Lastly, we were asked to rule out that UCHL1 features the most reactive catalytic cysteine within the different UCH DUBs. We would like to emphasize that the minimal probe GK16S

both in vitro and in cells binds / inhibits UCHL1 much more weakly than GK13S even though the cyanamide in both compounds possesses the same chemical reactivity. This stresses that chemical reactivity is not sufficient for the inhibition of UCHL1.

To experimentally address the reviewer's comment, we recombinantly expressed and purified UCHL3 wild-type as well as the A11I, L168F and A11I+L168F mutants (these were introduced in cellular experiments in Figure 6e). These mutations are in the Ubiquitin and GK13S recognition surfaces and thereby modulate the inhibitory potency of GK13S, but are not expected to change reactivity of the catalytic cysteine. Consequently, all four enzymes were reacting with / inhibited by the Cysteine-alkylating agent iodoacetamide with similar potency (i.e. have comparable intrinsic nucleophilic reactivity of the catalytic cysteine), but were inhibited with GK13S at different potencies. See new Supplementary Figures 10b-d.

We thank the reviewer for having raised this important point. Taken together, all these data conclusively show that GK13S is specifically recognized by UCHL1, and that this specificity is not the result of increased reactivity or abundance of UCHL1.

Free ubiquitin levels in glioblastoma cells

For this analysis (Figure 4F), it would be necessary to include a control where DJ-1 has been knocked down. Since the compounds are likely hitting DJ-1 as well under these conditions, it is important to demonstrate the reduction in free ubiquitin is solely due to UCHL1 inhibition.

We have repeated the experiment in the suggested PARK7/DJ-1-depleted background. We first tested that an efficient knock-down of PARK7 in U-87 MG cells is achieved (new Supplementary Figure 7f). We next found the reduction of monoubiquitin to only occur upon treatment with GK13S, but not with the PARK7-binding control probe GK16S, also in the PARK7-depleted background (new Supplementary Figure 7f).

Moreover, we also established that the knockdown of PARK7 itself does not change the levels of free monoubiquitin. This is shown with quantitation in new Supplementary Figure 7e.

The findings

- (i.) that free Ubiquitin levels are not changed by GK16S compared to treatment with DMSO in all three conditions tested (siScr, siUCHL3, siPARK7),
- (ii.) that GK16S leads to similar levels of PARK7-binding as does GK13S (see Figure 2 and Figure 4), and
- (iii.) and that the genetic depletion of PARK7 does not change monoubiquitin levels together demonstrate that the observed phenotype is due to inhibition of UCHL1 and that small-molecule inhibition of PARK7 does not change ubiquitin levels. These data demonstrate unequivocally that the reduction of free Ubiquitin is solely due to UCHL1 inhibition.

We fully agree with the reviewer that the compounds are hitting PARK7/DJ-1 as well under those conditions, however, we do not know the occupancy of cellular DJ-1 with the probes owing to the lack of a competitive probe (as we have it in Ub-VS for DUBs). As such, it will remain for future studies to evaluate how GK16S or related compounds can be used as specific (and potent/complete) inhibitors of DJ-1 in cells.

Crystallography

The maps don't quite look as well as one expects of a 2.24 Å structure. Is it due to diffraction anisotropy? Also, where did the ordered waters go? I did not see any modeled in the active site area next to the compound. There may be water-mediated interactions worth mentioning.

We fully agree with the reviewer that the map does not look at all like what one would expect from 2.24 Å, and the reviewer is right that this is the result of diffraction anisotropy. Table 1 lists resolution cut-offs of 2.24, 2.70 and even 3.22 Å along the three reciprocal axes, indicating that there is only a relatively small amount of high (>2.7 Å) resolution datapoints in the dataset. This high degree of anisotropy also explains why the quality of the map was so drastically improved by anisotropic scaling (compare to isotropic scaling to 2.6 Å). As a consequence, we have not had sufficient density to build many waters as the reviewer may also appreciate from inspecting the density (in line with a 2.7 or for sure a 3.2 Å map). The interpretation of water-like density around the compound is further complicated as the switching loop is disordered from residues 152-157 (consistent with the apo structure of UCHL1), and as such weakly defined density cannot unambiguously be interpreted as water molecules.

We have, however, closely inspected the 10 binding sites for potential water-mediated interactions between compound and protein but have not found any conclusive evidence. It is plausible that there is a bridging water molecule between the peripheral amide bond of the ligand and the side chain of Asn152, however, this could not be substantiated in the majority of copies since the rear part of the molecule covering this amide bond is poorly defined in all copies. All consistently observed contacts between ligand and protein are direct as shown in the figures.

In Supplementary Figure 6D, please zoom in close to the compound so the density envelope can be better appreciated.

We have improved the figure (now Supplementary Figure 8d) as suggested.

In the same figure, include a new digram showing overlay of key residues (Phe160, His161, etc) in different subunits. This may show protein dynamics in these residues that may explain inhibitor occupancy.

We thank the reviewer for this suggestion and have carried out the additional analysis which is shown in Supplementary Figure 9b and which demonstrates that the same binding mode is observed in all copies.

In addition, we have taken up a suggestion from a conference presentation where we were asked about the allosteric activation of UCHL1. New Supplementary Figure 9c shows that the hybrid conformation induced by the compound also includes the allosteric relay composed of Phe214, Phe53 and catalytic His161. This relay was described previously by Boudreaux et al. *PNAS* **2010**, *107*, 9117 and explains the allosteric activation of the catalytic triad of UCHL1 when the Leu8 loop of ubiquitin engages the S1 site. GK13S binding triggers the same conformational change.

These conformational state 'indicator' residues adopt the same conformation consistently across all 10 copies, suggesting that it is not really the "occupancy" of the inhibitor, which is different within the copies, but rather the degree of order. In line with this, the average *B* factors of the protein chains vary drastically from 46 Å² (chain H) to 77 Å² (chain I), which is also reflected in more difficult to interpret density in some copies. For this reason, we showed all copies and validated the interactions through extensive mutations which fully confirmed both the binding mode and the proposed specificity mechanism (Figures 5k and 6e).

Reviewer #2 (Remarks to the Author):

The manuscript by Malte Gersch and colleagues describes novel small molecule activity based probes and their structure-based specific interactions with UCHL1, a ubiquitin C-terminal hydrolase belonging to the deubiquitylating enzyme (DUB) family. UCHL1 has been of considerable interest for a long time as a potential target in neurodegeneration, but also cancer metastasis and invasiveness.

This study has been carried out with a very high standard in the quality of biochemical and structure-based experiments, and the characterisation of the lead compounds/probes to target UCHL1 as well as their molecular basis if specific binding has been carried out in a thorough manner. This work potentially reflects a nice contribution to the ubiquitin research community.

We thank the reviewer for this very positive assessment of our work.

However, before this work can be considered for publication, there are a number of points that should be addressed as outlined below.

Specific points:

1. Figure 1 and subsequent results indicate additional non-DUB targets for these compounds, such as PARK7 and C21orf33. What is the chemical basis for this potential cross-reactivity? For instance, PARK7 may have a glyoxalase/deglycase activity. The authors should discuss potential hydrolase mechanism of action and cross-reactivity as additional information may help with a further refinement of such probes in the future.

We fully agree with the reviewer that understanding the chemical basis for the observed cross-reactivity of 1,3-carboxycyanopyrrolidines and proteins of the PARK7 family is of high interest. We have therefore for a follow-up project solved crystal structures of human PARK7 in complex with GK16S and GK16R, showing the pyrrolidine sterically fitting into a very narrow pocket around its active site Cys-His dyad, but the amide not making any polar contacts. We would like to keep the focus of the current manuscript on UCHL1 and will follow up in detail in this direction in future studies.

The precise catalytic mechanism of PARK7 is intensely debated in the literature (see Choi et al. *FEBS J* **2014**, 281, 5447; Richarme et al. *Science* **2017**, 357, 208; Heremans et al. *PNAS* **2022**, 119, e2111338119.). C21orf33 (also named GATD3) has been annotated as a mitochondrial glutamine amidotransferase (an activity which is related to that of the also detected Omega-Amidase NIT2), but experimental evidence for its activity is scarce.

We have taken up the suggestion by the reviewer and included new Supplementary Figure 11 which shows three possible mechanisms for endogenous reactions catalyzed by PARK7 and C21orf33. We contrast these with reactions with cyanamide compounds. All these mechanisms involve a nucleophilic attack into an electrophilic center. In the case of PARK7, the selection seems to be based on the relatively small size of the electrophile as no other chemical resemblances became apparent.

In the case of glutamine processing by C21orf33, however, one can see a chemical resemblance to a 1,3-carboxycyanopyrrolidine which provides a rational for the observed cross-reactivity. The carbon which is attacked by the C21orf33 catalytic cysteine in both glutamine and the cyanamide is five positions away from a carbonyl (carboxylate in glutamine, amide in the cyanamide) which allows for the possibility that the 1,3-carboxypyrrolidine cyanamide is recognized not only purely by reactivity.

We thank the reviewer for suggesting this analysis which we have added to the discussion section of the manuscript.

2. This is not the first report describing small molecular probes & inhibitors against UCH-L1. A comparison of GK to previously characterised inhibitors and probes, such as isatin O-acyl oxime compound (LDN-5744) (Liu et al., *Chem Biol.* 2003 Sep;10(9):837-46.) or IMP-1710 (Nattawadee et al.,

J. Am. Chem. Soc. 2020, 142, 28, 12020–12026) would be very insightful, in particular when describing the molecular basis for selectivity.

We thank the reviewer for the suggestion to extend the comparison to previously described compounds. Regarding LDN-57444, we had written already:

“We included the isatin O-acyl oxime LDN-574444 which has been widely used as a specific UCHL1 inhibitor, whose effectiveness, however, has recently been questioned.ref26,28 We found that GK13S, but neither GK16S nor LDN-57444, led to inhibition of cellular UCHL1 in U-87 MG cells (Fig. 4d).”

This experiment showed that LDN-57444, which was introduced in 2003 as one of the earliest DUB inhibitors, does not lead to inhibition of cellular UCHL1 as detectable by Ub-VS competition, and is in agreement with data from the referenced other groups where the compound neither inhibited cellular nor recombinant UCHL1. Overall our data strengthens the case for GK13S as a non-toxic and effective UCHL1 inhibitor.

We have not included IMP-1710, however, we have included compounds Cpd117 and CG173 which are highly similar (see Figure 1). While not mentioned in the manuscript, we made the parent inhibitor of IMP-1710 (so IMP-1710 lacking the alkyne) which behaves comparable to compounds Cpd117 and CG173. Importantly, all these 1,2-carboxycyanopyrrolidines show potent inhibition of recombinant UCHL1 (including hyper-reactivity, i.e. covalent reaction with more than one molecule probe per enzyme), however are not usable in cellular experiments due to their off-target toxicity (Supplementary Figures 5, 6, and 7).

We had already included in the discussion:

„This demonstrates that the high toxicity of the 2-carboxy-N-cyanopyrrolidine CG173 is unrelated to UCHL1.“

We have taken up the suggestion of the reviewer to make this point more clear as this effectively calls the use of 1,2-linked carboxycyanopyrrolidines as UCHL1 inhibitors into question. Panyain et al. *J Am Chem Soc* **2020**, 142, 28, 12020–12026 do not show sustained cellular inhibition of UCHL1 by IMP-1710 (e.g. after 24h, see our Figures 3e/4d) and the studied phenotype is immediate cytotoxicity which as described in our manuscript does not fit to what one would expect from genetic perturbation and what we see from full inhibition of UCHL1 with GK13S. However, it is possible that there are cell-line specific and possibly compound-batch specific issues that need to be considered, which is why we think an experimental thorough side-by-side comparison should be part of a future study.

As we have strengthened in the discussion, we are excited to here present the first non-toxic inhibitor of UCHL1 which shows complete and sustained inhibition of the cellular enzyme, phenocopies the UCHL1-mutant mice and can be used in a pair of probes to investigate specifically the function of UCHL1. It is also the first compound for which the basis for UCHL1 specificity has been revealed. This combination of properties makes the advance of our work over previously reported compounds clear.

3. Proteomics data used in the present study (method described in lines 1075-1137) is recommended to be deposited into a public repository (e.g. PRIDE) or similar.

All proteomics data have been deposited into the ProteomeXchange repository. This was previously apparent only from the reporting summary. We have added a data availability paragraph to the main text stating:

“Data have been deposited with the protein data bank under accession code 7ZM0, and with ProteomeXchange under accession codes MSV000090044 and MSV000090045. Chemical characterization data as well as uncropped gels and blots are provided in the Supplementary Information. All data are available on request from the authors.”

Reviewer #1 (Remarks to the Author):

In this revised manuscript, the authors have made a sincere effort in addressing each and every point brought up during the review of the original version of the manuscript. Their response has improved certain specific areas and the overall quality of this paper, which was already a very good one to start with. Replacing the older proteomics results with a more carefully and controlled analysis has yielded a better sense of the spectrum of cellular targets of GK13S. I believe their effort has resulted in a version that can be published as such.

Reviewer #1 (Remarks to the Author):

In this revised manuscript, the authors have made a sincere effort in addressing each and every point brought up during the review of the original version of the manuscript. Their response has improved certain specific areas and the overall quality of this paper, which was already a very good one to start with. Replacing the older proteomics results with a more carefully and controlled analysis has yielded a better sense of the spectrum of cellular targets of GK13S. I believe their effort has resulted in a version that can be published as such.

We are very grateful to the reviewer for appreciating our revision and for supporting publication of this work.